# TEST-REAL-TIME ADAPTATION AGAINST SPARSE KNOWLEDGE BOTTLENECK

## ABSTRACT

Test-time adaptation (TTA) typically involves adaptation delays due to self-training, which conflict with real-time deployment where inference cannot pause for adaptation. We introduce Test-Real-Time Adaptation (TRTA), which requires uninterrupted prediction while adaptation runs in the background, leaving few update opportunities. In TTA, later reliable signals enable error correction and steady knowledge accumulation, whereas in TRTA, such signals are rare, so knowledge growth stalls. We term this the sparse-knowledge bottleneck, where limited updates hinder error correction and increase the risk in self-training. To solve this challenge, we propose a novel method, dubbed Agreement- and Uncertainty-Guided Reweighting (AUGR). AUGR fuses two complementary evidence sources: *(i)* inter-model agreement, defined as the concordance of predicted class rankings between the base and the reference models on each sample, revealing common knowledge with consensus predictions; and *(ii)* inner-model uncertainty, representing the reliability of such knowledge, which balance the agreement evidence by discounting low-confidence cases. By integrating both sources of evidence, AUGR emphasizes the learning of consistent, reliable samples and suppresses conflicting or uncertain ones, thereby promoting robust knowledge accumulation. Extensive experiments on ImageNet-C/R/K demonstrate the effectiveness of AUGR combating sparse-knowledge bottleneck in TRTA. Code will be released.

## 1 INTRODUCTION

Test-time adaptation (TTA) (Wang et al., 2021; 2022) adapts a pre-trained model to distribution shifts during inference, typically following a **predict-then-adapt** routine. This sequential scheme requires pausing inference between consecutive inputs, which undermines real-time responsiveness. Such latency is unacceptable in many practical applications, such as autonomous driving (Hu et al., 2023) or online video analytics (Chen et al., 2024b), where every frame must be processed without delay. To address this limitation, we consider a real-time variant of TTA, termed **Test-Real-Time Adaptation (TRTA)**, which follows a **predict-and-adapt** routine, meaning that prediction for the current input and adaptation of the model proceed simultaneously without synchronization pauses. As illustrated in Fig. 1, TRTA (c) eliminates the waiting overhead of TTA (a), while retaining adaptability absent in direct inference (b). Its throughput thus matches direct inference but with online updates, making TRTA more suitable for time-critical scenarios.

In practice, adaptation typically requires several times more computation than forward inference, for example due to backpropagation steps. As a result, under TRTA the number of model updates is much smaller than the number of predictions, leading to a sparse adaptation regime. This mismatch severely limits the model's capacity to correct early mistakes and creates what we call a *sparse-knowledge bottleneck*. Consequently, applying conventional TTA methods in this setting becomes inherently inadequate because they rely on frequent updates to accumulate reliable pseudo-labels over time. In Fig. 2, we track predictive mutual information (PMI) on target-stream outputs, which captures simultaneous uncertainty reduction and preservation of label-space diversity (Tishby et al., 2000; Houlsby et al., 2011; Kendall & Gal, 2017). The results show that PMI grows steadily under TTA, reflecting continuous knowledge acquisition, whereas under TRTA the curve is significantly compressed and often flattens, which indicates stalled learning.

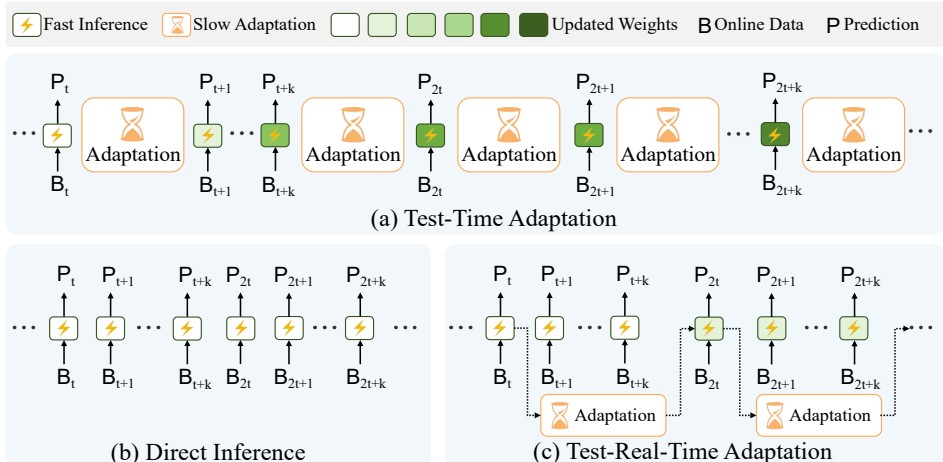

Figure 1: TTA: **predict-then-adapt** with adaptation pauses between inferences. DI: prediction only. TRTA: **predict-and-adapt**, inference runs without pauses while adaptation runs concurrently.

This observations highlight that a central challenge of TRTA is how to extract and preserve useful domain knowledge when adaptation opportunities are extremely sparse. One possible way to mitigate this bottleneck is to design faster adaptation routines so that more updates can be executed, but this often reduces compatibility with existing methods. *In this work, we instead focus on improving the effectiveness of existing TTA algorithms under nearly the same efficiency, enabling them to adapt better to TRTA with only simple modifications.*

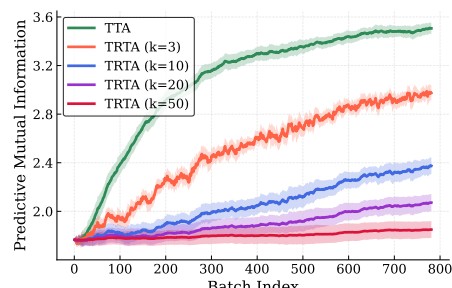

Figure 2: Changing trends in predictive mutual information on ImageNet-C (Gaussian Noise), where $k$ denotes adaptation delay.

Sparse updates in TRTA make parameter adaptation highly vulnerable to noise, since each update has disproportionate influence on the model trajectory. To counteract this, the model must prioritize signals that reflect common knowledge, which are generally robust against individual noisy samples (Li & Hoiem, 2017; Zhang et al., 2018; Han et al., 2018; Zhu & Li, 2021). Building on this, we propose **A**greement- and **U**ncertainty-**G**uided **R**eweighting (**AUGR**). AUGR leverages a reference model (e.g., a historical or foundation model) and measures its consensus with the base model, integrating two complementary evidence sources: *(i) inter-model agreement*, defined as the concordance of predicted class rankings between the base and reference models, which highlights knowledge consistently supported by both; and *(ii) inner-model uncertainty*, which evaluates the reliability of such agreement by down-weighting low-confidence predictions. By combining agreement as a proxy for common knowledge with uncertainty as a safeguard against unreliable cases, AUGR selectively emphasizes stable, trustworthy samples and suppresses conflicting or ambiguous ones. This design directly addresses the sparse-knowledge bottleneck by ensuring that each limited update contributes to robust knowledge accumulation. Extensive experiments demonstrate the effectiveness of AUGR, e.g., a 9.03% , 3.88% , and 5.50% improvement on ImageNet-C/R/K, respectively. Moreover, AUGR shows flexibility as a plug-and-play module, enabling existing methods to operate under TRTA.

The contributions of this work can be summarized as follows:

(1) We propose a new Test-Real-Time Adaptation (TRTA) paradigm, which align with the timeliness requirements of real-world applications.

(2) We reveal a new challenge in TRTA, i.e., the sparse knowledge bottleneck, and propose a novel Agreement- and Uncertainty-Guided Reweighting (AUGR) method to mitigate this challenge from converging evidence from multiple sources.

(3) AUGR demonstrates effectiveness across various benchmarks and shows flexibility as a plug-and-play module, enabling existing methods to operate under TRTA.

## 2 RELATED WORK

**Test-Time Adaptation.** Test-time adaptation (TTA) mitigates performance degradation under source–target distribution shifts by adapting a pre-trained model with only unlabeled online data. Existing works have demonstrated strong success through diverse self-training techniques, including sample selection and reweighting (Niu et al., 2022; Lee et al., 2024; Marsden et al., 2024), uncertainty estimation (Zhang et al., 2025), sample augmentation with consistency regularization (Zhang et al., 2022; Wang et al., 2022; Chen et al., 2022), cross-model co-learning (Chen et al., 2024c), and sharpness-aware optimization (Niu et al., 2023). Although several works have studied efficient TTA, they typically reduce cost by performing backpropagation on only a subset of data (Chen et al., 2024c;a) or by selectively skipping adaptation steps (Colomer et al., 2023; Chen et al., 2024a), while still adhering to a predict-then-adapt workflow. Despite their efficacy, their inevitable adaptation delays conflict with the timeliness requirements of real-world applications. In contrast, TRTA differs a parallel paradigm, as it operates under asynchrony between inference and adaptation.

**Asynchronous Inference and Training.** Real-time systems (Tosi et al., 2024) favor a predict-and-update paradigm, where inference continues while parameters are updated in the background; this pattern is widely adopted in edge–cloud collaborative learning (Gan et al., 2023; Zhuang et al., 2024; Li et al., 2025), online continual learning (Ghunaim et al., 2023), and federated learning (Duan et al., 2022). However, these systems typically assume ample labeled source/target data for supervision, whereas TTA operates without labels at test time (and often without access to source data). Unlike those real-time systems, we study a more challenging test-real-time adaptation (TRTA) setting that couples real-time constraints with label-free adaptation, serving as a complementary regime to prior real-time work.

## 3 METHOD

### 3.1 PROBLEM DEFINITION

Both TTA and TRTA consist of two objectives: prediction and adaptation. The key difference lies in how these two steps are scheduled. TTA follows a *predict-then-adapt* routine, where inference is paused until adaptation is completed. In contrast, TRTA enforces a *predict-and-adapt* routine, where inference and adaptation proceed in parallel without synchronization waits. Formally, let $F_\Theta$ denote the base model, and let $\mathcal{X}_t$ be the $t$-th incoming online batch. The model produces predictions

$$\hat{y}(\mathcal{X}_t) = \arg\max(\mathbf{p}(\mathcal{X}_t)) , \tag{1}$$

where $\mathbf{p}(\mathcal{X}) = \mathrm{softmax}(F_\Theta(\mathcal{X}))$ is the softmax output logits. The parameters $\Theta$ are updated using self-supervised loss objectives, such as entropy loss:

$$\mathcal{H}(\mathcal{X}_t) = - \mathbf{p}(\mathcal{X}_t) \log \mathbf{p}(\mathcal{X}_t) . \tag{2}$$

To ensure real-time inference, TRTA introduces a parallel inference model $F_\Omega$, initialized with $\Omega \leftarrow \Theta_0$, which continuously serves predictions on the input stream. Let $k$ denote the adaptation delay measured in batches, i.e., the number of subsequent inference batches processed while that adaptation runs. As shown in Fig. 1 (c), the parameters of $F_\Omega$ are synchronized with $F_\Theta$ according to

$$\Omega = \begin{cases} \Theta_n, & \text{if the } n\text{-th adaptation has completed,} \\ \Theta_{n-1}, & \text{otherwise,} \end{cases} \tag{3}$$

where $n \in \mathbb{N}^+$ indexes completed adaptations, and each adaptation requires approximately $k$ time units to finish. In this way, $F_\Omega$ always continues serving predictions without waiting, and switches to the newly adapted parameters once $F_\Theta$ finishes its update.

This formulation highlights the structural difference between TTA and TRTA. TTA performs *dense updates* (i.e., one update per batch), enabling subsequent reliable pseudo-labels to correct earlier mistakes. TRTA, however, performs *sparse updates* due to adaptation delays $k$, leaving $F_\Omega$ to infer with outdated parameters and $F_\Theta$ to adapt with limited feedback. This mismatch restricts knowledge accumulation and results in the *sparse-knowledge bottleneck*, motivating the methods we propose in the following sections.

## 3.2 Agreement and Uncertainty-Guided Reweighting

Sparse updates in TRTA make adaptation highly sensitive to noise, since each erroneous pseudo-label may persist for a long time with few opportunities for correction. To address this issue, we propose Agreement- and Uncertainty-Guided Reweighting (**AUGR**), which constructs reliable supervision signals by combining two complementary sources of evidence. First, *agreement-guided reweighting* extracts inter-model agreement between the base and reference models to capture common knowledge. Second, *uncertainty-guided reweighting* calibrates this consensus by evaluating prediction confidence, reducing the impact of unreliable agreements. Finally, the two signals are fused into a unified reweighting factor that drives both adaptation and knowledge distillation losses. We describe each component in detail below.

### 3.2.1 Agreement-Guided Reweighting.

Under TRTA, sparse updates leave the model highly sensitive to noisy signals, since early errors cannot be promptly corrected. To reduce error propagation, it is crucial to emphasize robust knowledge rather than noise-prone cues. Such knowledge is difficult to identify with a single predictor, but can be more reliably extracted by exploiting agreement across models with distinct parameter states (Li & Hoiem, 2017; Zhang et al., 2018; Han et al., 2018; Zhu & Li, 2021), such as foundation or historical models. Inspired by this, we introduce a reference model $G_\Phi$ alongside the base model $F_\Theta$ and use their consensus to elevate reliable supervision and suppress noisy predictions.

Specifically, let the softmax output logits of the base and reference models be denoted as $\mathbf{p}_b$ and $\mathbf{p}_r$, respectively. For each pair of predictions, we first extract their respective TopK predicted classes as:

$$Z_b = \text{TopK}(\mathbf{p}_b), \ Z_r = \text{TopK}(\mathbf{p}_r), \tag{4}$$

where $Z \in \mathbb{R}^{B \times C}$ and $B$ is the batch size. For classes appearing in both sets, $M = Z_b \cap Z_r$, reliable consensus should not only predict the same label but also rank it similarly. We therefore compute a rank-alignment score:

$$\text{score}_{\text{align}}(\mathbf{p}_b, \mathbf{p}_r) = \frac{1}{|M|} \sum_{c \in M} \left(1 - \frac{||\text{rank}_{Z_b}(c) - \text{rank}_{Z_r}(c)||}{\text{K} - 1 + \epsilon}\right), \tag{5}$$

where $\epsilon$ is a small constant for numerical stability. $\text{rank}_{Z_b}(c)$ and $\text{rank}_{Z_r}(c)$ denote the positions of class $c$ in the TopK predictions of the base and reference models, respectively.

However, rank alignment alone cannot capture cases where the two TopK sets barely overlap. For instance, if the base model predicts "animals" while the reference predicts "vehicles", their rank consistency within the tiny intersection may be high, but the overall disagreement is still large. To penalize such mismatches, we further compute the Jaccard index:

$$\text{score}_{\text{Jaccard}}(\mathbf{p}_b, \mathbf{p}_r) = \frac{|Z_b \cap Z_r|}{|Z_b \cup Z_r|}. \tag{6}$$

This term measures the degree of overlap between the two TopK sets. A small or empty intersection yields a low Jaccard score, thereby down-weighting samples where the two models disagree strongly.

The final agreement score is obtained by a linear mixup of rank alignment and set overlap, where a coefficient $\alpha \in [0, 1]$ balances their contributions:

$$\text{score}_{\text{a}}(\mathbf{p}_b, \mathbf{p}_r) = \alpha \cdot \text{score}_{\text{align}}(\mathbf{p}_b, \mathbf{p}_r) + (1 - \alpha) \cdot \text{score}_{\text{Jaccard}}(\mathbf{p}_b, \mathbf{p}_r). \tag{7}$$

By combining rank alignment, which captures fine-grained consistency within overlapping predictions, with Jaccard overlap, which measures the breadth of shared evidence, the agreement score provides a balanced and reliable estimate of inter-model agreement. This calibrated signal ensures that only strong and broad agreements are emphasized, thereby suppressing spurious correlations and helping the model accumulate stable knowledge under TRTA's sparse update regime.

### 3.2.2 Uncertainty-Guided Reweighting

While agreement offers a useful consensus signal, it does not by itself guarantee correctness. Two models may still agree on an incorrect label or disagree on a correct one. As shown in Fig. 3, predictions can be visualized on a normalized agreement–certainty grid, where color encodes correctness. This reveals two critical cases: (i) strong agreement accompanied by high entropy (Area 4) versus low-entropy agreement (Area 2), and (ii) weak agreement accompanied by low-entropy (Area 1) versus high-entropy disagreement (Area 3). Empirically, samples in Area 4 have lower mean accuracy than those in Area 2 and should not be weighted equally, whereas samples in Area 1 are more accurate than those in Area 3 and deserve greater emphasis.

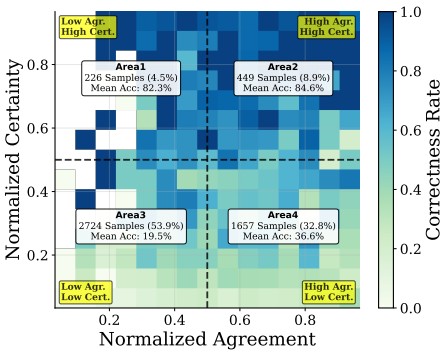

Figure 3: Complementarity of AGR & UGR.

Therefore we complement agreement with uncertainty to better separate reliable from noisy evidence. Low-entropy samples are likely to be correct even with modest agreement, while high-entropy samples remain unreliable even with strong agreement. To capture this, we define an entropy-based weight

$$\text{score}_{\text{u}}(\mathbf{p}) = \frac{1}{\exp\{\mathcal{H}(\mathbf{p}) - \text{m}\}} \, , \qquad (8)$$

where $\mathcal{H}$ denotes the entropy and m is a pre-defined margin. Eq. (8) defines an exponential weighting function anchored at m. Predictions with lower entropy receive larger weights, while predictions with higher entropy are progressively downweighted.

**Discussion**. The entropy-based weighting in Eq. (8) allows us to softly adjust the contribution of each sample according to its confidence. However, existing uncertainty-based reweighting strategies in TTA (Niu et al., 2022; 2023; Lee et al., 2024) adopt a hard selection scheme that discards high-entropy samples and reweights only those retained. While effective under TTA, this strategy becomes problematic in TRTA. As shown in Fig. 4, samples are partitioned by entropy using margin m and by pseudo-label correctness, and each cell reports its share of the population. From (a) TTA to (b) TRTA, the proportion of low-entropy true positives (Area 1) decreases, and false positives

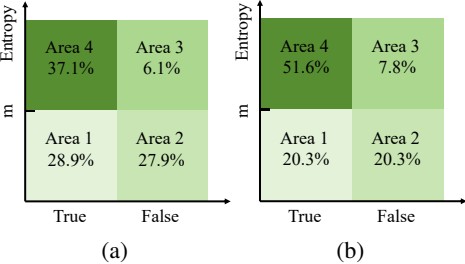

Figure 4: (a) and (b) present the sample distributions across entropy and label correctness in TTA and TRTA; each block shows its proportion among all adapted samples.

(Area 2) take a larger share among the low-entropy group. At the same time, true but high-entropy samples (Area 4) increase but are discarded since sparse knowledge prevents their entropy from dropping below the margin. These shifts show that directly reusing uncertainty-based reweighting under TRTA fails to preserve enough reliable supervision. In contrast, UGR replaces hard gating with soft per-sample weighting, which enables fuller use of informative samples and strengthens the consensus signal to provide more reliable guidance. Empirical results (Tables 1–3) confirm the effectiveness of this design and highlight its necessity for adapting existing methods to TRTA across diverse distribution shifts.

### 3.2.3 AGREEMENT AND UNCERTAINTY-GUIDED REWEIGHTING

We combine the agreement and uncertainty scores into a unified reweighting factor (AUGR):

$$w_\pi = \lambda_{\text{a}} \, \text{score}_{\text{a}}(\mathbf{p}_b, \mathbf{p}_r) + \lambda_{\text{u}} \, \text{score}_{\text{u}}(\mathbf{p}_\pi), \qquad \pi \in \{b, r\}, \qquad (9)$$

where $\lambda_{\text{a}}$ and $\lambda_{\text{u}}$ control the scale of agreement and uncertainty scores, respectively. We apply AUGR to the widely used entropy minimization and further incorporate the cross-entropy loss $\mathcal{L}_{\text{CE}}$ for knowledge distillation, formulated as:

$$\mathcal{L}(\mathcal{X}) = \begin{cases} w_{\text{b}} \cdot \mathcal{H}_b(\mathcal{X}) \; + \; w_b \cdot \mathcal{L}_{\text{CE}}(\mathbf{p}_b(\mathcal{X}) \; ; \; \hat{y}_r(\mathcal{X})), & \text{for the base model,} \\ w_r \cdot \mathcal{H}_r(\mathcal{X}), & \text{for the reference model,} \end{cases} \qquad (10)$$

For clarity of our design, we summarize TRTA and AUGR in Algorithms 1 and 2, respectively. The inference model continuously predicts online data using the most recently updated weights

**Algorithm 1** TRTA

**Inputs:** Inference Model: $I_\Omega$, $\Omega \leftarrow \Theta_0$
**Inputs:** `Global` *Ready* $\leftarrow$ **True**, *Num* $n \leftarrow 0$
**Inputs:** Training Batches: $\{\mathcal{X}_1, \mathcal{X}_2, \cdots, \mathcal{X}_T\}$
1: **for** $t \leftarrow 1$ to $T$ **do** *(Main)*
2:     Get Predictions via Eq. (1), Check *Ready*
3:     **if** *Ready* **then** *(Parallel)*
4:         $\Theta_n \leftarrow \mathbf{AUGR}(\mathcal{X}_t)$, $\Omega \leftarrow \Theta_n$, $n \leftarrow n+1$
5:     **end if**
6: **end for**
**Outputs:** Predictions

**Algorithm 2** AUGR

**Inputs:** Base & Reference Models: $F_\Theta$, $G_\Phi$
**Inputs:** $\Theta \leftarrow \Theta_0$, $\Phi \leftarrow \Phi_0$
**Inputs:** Training Batch: $\mathcal{X}_t$
1: Set *Ready* $\leftarrow$ **False**
2: Compute AUGR scores using Eq. (5)-(9)
3: Update the base and reference models using Eq. (10)
4: Get $\Theta_n$
5: Set *Ready* $\leftarrow$ **True**
**Outputs:** $\Theta_n$

and switches to new weights as soon as they are ready. The inference and adaptation procedure is parallel, forming a predict-and-adapt workflow. In summary, agreement-guided reweighting exploits cross-model agreement, and uncertainty-guided reweighting further complements it to prevent low-confidence predictions from dominating. Each strategy is effective on its own, yet their combination yields further improvements (see Table 5). AUGR integrates both to provide reliable weighting signals for self-training, thereby mitigating the sparse-knowledge bottleneck.

# 4 EXPERIMENTS

## 4.1 EXPERIMENTAL SETUP

**Benchmarks.** To adequately evaluate our method, we conduct experiments on popular test-time adaptation benchmarks covering both synthetic (ImageNet-C (Hendrycks & Dietterich, 2019)) and real-word distribution shifts (ImageNet-R (Hendrycks et al., 2021) and ImageNet-K (Wang et al., 2019)). See details in Appendix B.

**Baselines.** We compare our method with state-of-the-art (SOTA) methods, including Tent (Wang et al., 2021), ETA (Niu et al., 2022), SAR (Niu et al., 2022), COME (Zhang et al., 2025) and CEMA (Chen et al., 2024c). We equip the cross-entropy loss for Tent, ETA, SAR and COME to distillate knowledge from the reference model for fair comparison. See details in Appendix C.

**Implementation Details.** By default, we set the architecture of the base and reference models as ResNet-50 (He et al., 2016) and VitBase (Dosovitskiy et al., 2021), respectively. And we adopt the official checkpoints from `Pytorch` library and `timm` repository following existing methods (Niu et al., 2023). We use SGD optimizers with a learning rate of 0.00025 and 0.001 with a momentum of 0.9 for collaborative models and tune the parameters of their normalization layers. The batch size is set to 64. For hyper-parameters, we set the TopK in Eq. (4), the coefficient $\alpha$ in Eq. (7), the entropy margin m in Eq. (8), the scale factors $\lambda_a$ and $\lambda_u$ in Eq. (9) as 50, 0.9, $0.4 \times \log(C)$, 5, and 3, respectively. We report the average accuracy across 5 random seeds. All experiments are conducted on a server with NVIDIA 3090 GPU and Intel(R) Xeon(R) Gold 6330 CPU. See details in Appendix C.

## 4.2 COMPARISON WITH SOTAS

We present results on ImageNet-C (severity level 5) and ImageNet-R/K in Table 1 and Table 2, respectively. Both the classification accuracy and the number of model updates are reported to comprehensively assess the performance of the baseline methods. In these tables, ✓ indicates the performance when combined with our method. By default, we set the base and reference models as ResNet-50 and VitBase, respectively.

**Performance on ImageNet-C/R/K.** From Table 1, we have the key observations:
**(a)** Adaptation delay plays a critical role in TRTA, as higher efficiency in each adaptation step allows more frequent updates and generally yields better performance. For example, SAR requires more time per update than other methods because each adaptation step involves two backward passes, which reduces its overall update times. This exacerbates the sparse knowledge bottleneck and ultimately degrades its performance.

Table 1: Accuracy comparisons in TRTA on ImageNet-C.

| Methods | AUGR | Gauss. | Shot | Impul. | Defoc. | Glass | Motion | Zoom | Snow | Frost | Fog | Brit. | Contr. | Elast. | Pixel | JPEG | AVG | #Updates |
|---|---|---|---|---|---|---|---|---|---|---|---|---|---|---|---|---|---|---|
| ResNet-50 | | 2.21 | 2.93 | 1.85 | 17.92 | 9.82 | 14.78 | 22.49 | 16.89 | 23.30 | 24.43 | 58.93 | 5.43 | 16.95 | 20.60 | 31.64 | 18.01 | 0 |
| VitBase | | 46.86 | 47.59 | 46.88 | 42.73 | 34.21 | 50.45 | 44.70 | 56.87 | 52.63 | 56.52 | 76.06 | 31.79 | 46.72 | 65.49 | 66.03 | 51.04 | 0 |
| Tent | | 21.88 | 22.92 | 22.20 | 21.32 | 20.76 | 33.18 | 42.79 | 40.61 | 38.03 | 52.15 | 66.37 | 24.67 | 47.82 | 53.75 | 46.42 | 36.99 | 79 |
| | ✓ | 34.14 | 36.16 | 35.45 | 32.16 | 32.43 | 44.44 | 49.13 | 49.58 | 44.77 | 57.85 | 66.83 | 39.97 | 54.65 | 59.04 | 53.67 | 46.02 | 79 |
| | △ | 12.26↑ | 13.24↑ | 13.25↑ | 10.84↑ | 11.67↑ | 11.26↑ | 6.34↑ | 8.97↑ | 6.74↑ | 5.70↑ | 0.46↑ | 15.30↑ | 6.83↑ | 5.29↑ | 7.25↑ | 9.03↑ | 0 |
| SAR | | 16.40 | 18.01 | 17.80 | 16.75 | 17.45 | 28.26 | 40.42 | 36.03 | 34.78 | 48.99 | 65.95 | 13.95 | 45.31 | 50.36 | 41.87 | 32.82 | 36 |
| | ✓ | 24.59 | 28.59 | 26.63 | 23.00 | 24.75 | 36.09 | 45.84 | 43.42 | 39.93 | 54.40 | 66.70 | 24.50 | 51.02 | 55.65 | 48.85 | 39.60 | 36 |
| | △ | 8.19↑ | 10.58↑ | 8.83↑ | 6.25↑ | 7.30↑ | 7.83↑ | 5.42↑ | 7.39↑ | 5.15↑ | 5.41↑ | 0.75↑ | 10.55↑ | 5.71↑ | 5.29↑ | 6.98↑ | 6.78↑ | 0 |
| COME | | 24.03 | 26.16 | 24.74 | 22.79 | 23.11 | 36.58 | 45.38 | 43.75 | 40.36 | 54.04 | 66.93 | 29.80 | 50.58 | 55.82 | 48.70 | 39.52 | 79 |
| | ✓ | 33.30 | 35.09 | 33.67 | 31.41 | 31.21 | 43.96 | 49.22 | 49.12 | 44.34 | 57.66 | 67.62 | 40.18 | 54.42 | 59.15 | 53.16 | 45.57 | 79 |
| | △ | 9.27↑ | 8.93↑ | 8.93↑ | 8.62↑ | 8.10↑ | 7.38↑ | 3.84↑ | 5.37↑ | 3.98↑ | 3.62↑ | 0.69↑ | 10.38↑ | 3.84↑ | 3.33↑ | 4.46↑ | 6.05↑ | 0 |
| ETA | | 25.57 | 28.11 | 26.73 | 24.01 | 24.78 | 37.42 | 46.40 | 44.47 | 40.99 | 54.95 | 67.07 | 30.76 | 51.62 | 56.46 | 49.42 | 40.58 | 79 |
| | ✓ | 33.37 | 35.25 | 33.85 | 31.44 | 31.17 | 43.72 | 49.15 | 49.26 | 44.39 | 57.67 | 67.47 | 39.72 | 54.59 | 59.03 | 53.04 | 45.54 | 79 |
| | △ | 7.80↑ | 7.14↑ | 7.12↑ | 7.43↑ | 6.39↑ | 6.30↑ | 2.75↑ | 4.79↑ | 3.40↑ | 2.72↑ | 0.40↑ | 8.96↑ | 2.97↑ | 2.57↑ | 3.62↑ | 4.96↑ | 0 |
| CEMA | | 24.06 | 34.00 | 33.68 | 21.64 | 25.73 | 37.02 | 46.63 | 42.84 | 41.17 | 51.77 | 64.88 | 19.13 | 50.54 | 55.42 | 50.45 | 39.93 | 66 |
| | ✓ | 29.28 | 36.36 | 34.89 | 26.78 | 28.47 | 40.72 | 47.32 | 45.60 | 42.26 | 54.29 | 65.47 | 33.50 | 51.44 | 56.56 | 51.69 | 42.98 | 49 |
| | △ | 5.22↑ | 2.36↑ | 1.21↑ | 5.14↑ | 2.74↑ | 3.70↑ | 0.69↑ | 2.76↑ | 1.09↑ | 2.52↑ | 0.59↑ | 14.37↑ | 0.90↑ | 1.14↑ | 1.24↑ | 3.05↑ | 17↓ |

Table 2: Accuracy comparisons on ImageNet-R/K. "X(Y)" denotes Acc(#Updates).

| | AUGR | Tent | SAR | COME | ETA | CEMA |
|---|---|---|---|---|---|---|
| IN-R | | 41.83 (94) | 40.16 (53) | 43.19 (94) | 43.31 (94) | 46.36 (94) |
| | ✓ | 45.71 (94) | 42.09 (53) | 46.60 (94) | 46.69 (94) | 47.63 (79) |
| | △ | 3.88↑ (0) | 1.93↑ (0) | 3.41↑ (0) | 3.38↑ (0) | 1.27↑ (15↓) |
| IN-K | | 31.78 (199) | 27.89 (133) | 34.12 (199) | 34.05 (199) | 36.73 (199) |
| | ✓ | 37.28 (199) | 32.82 (133) | 38.01 (199) | 38.01 (199) | 37.08 (160) |
| | △ | 5.50↑ (0) | 4.93↑ (0) | 3.89↑ (0) | 3.96↑ (0) | 0.35↑ (39↓) |

Table 3: Accuracy Comparisons on mixed distribution shifts.

| | AUGR | Tent | SAR | COME | ETA | CEMA |
|---|---|---|---|---|---|---|
| S-5 | | 36.08 | 31.95 | 37.31 | 38.35 | 47.11 |
| | ✓ | 41.87 | 37.05 | 43.69 | 43.41 | 47.21 |
| | △ | 5.79↑ | 5.10↑ | 6.38↑ | 4.96↑ | 0.10↑ |
| S-3 | | 55.68 | 56.48 | 55.10 | 52.12 | 61.85 |
| | ✓ | 60.26 | 60.72 | 60.56 | 56.05 | 62.18 |
| | △ | 4.58↑ | 4.24↑ | 5.46↑ | 3.93↑ | 0.33↑ |

**(b)** Existing methods only partially alleviate the sparse knowledge bottleneck. In TRTA, noisy self-supervised signals cannot be easily corrected due to the scarcity of subsequent reliable ones. As illustrated in Fig. 4, they lack enough knowledge to lower the entropy of data below the margin, and consequently fails to produce more reliable self-supervised signals. As a result, the bottleneck becomes more pronounced, further undermining the effectiveness of prior TTA methods.

**(c)** It is possible to accumulate robust knowledge even with fewer updates. For example, CEMA achieves competitive performance with only 66 updates by training the model on buffered samples, where reliable data are retained. This strategy mitigates the impact of noisy signals from new data and provides a viable way to accumulate robust knowledge in TRTA. However, its effectiveness remains limited, since the retained samples capture only a partial view of the target domain.

**(d)** Our AUGR method consistently improves existing methods by a large margin. Specifically, AUGR mitigates the sparse knowledge bottleneck by promoting reliable common knowledge while suppressing noisy uncommon knowledge, thereby strengthening self-training. The results in Table 2 further validate the effectiveness of our method, showing that it generalizes well beyond synthetic noise to real-world domain shifts.

**Performance under mixed distribution shifts.** In real-world applications, incoming data may exhibit heterogeneous distribution shifts (Niu et al., 2023; Yuan et al., 2023). To assess robustness to such heterogeneity, we evaluate the comparison methods on a mixture of 15 corruption types at severity levels 5 and 3, with results summarized in Table 3. For reference, the standalone performance of the base and reference models is reported in Table 4. From Table 3, we observe that AUGR consistently improves the baselines, with an average gain of 3.47% and 3.71% at severity levels 5 and 3, respectively. For scenarios with extremely few updates, such as SAR (Niu et al., 2023), AUGR still alleviates the sparse-knowledge bottleneck and delivers stable gains. CEMA (Chen et al., 2024c) shows excellent performance on this task, probably profits from its additional buffer that contain reliable samples.

**Performance under online imbalanced label distribution shifts.** We consider a challenging real-world streaming setting in which incoming labels exhibit temporal correlation. Following prior works (Niu et al., 2023; Chen et al., 2024c), we use a ResNet-50 with Group Normalization (GN) as the base model to avoid collapse caused by biased batch-normalization statistics. The results are summarized in Table 4, we omit the number of updates since they are identical as in Table 1. We observe that baseline methods degrade more under imbalanced label-distribution shifts than under balanced ones, primarily because each update sees only a subset of classes, which exacerbates

Table 4: Accuracy comparisons in TRTA on ImageNet-C (online imbalanced label distribution shifts).

| Methods | AUGR | Gauss. | Shot | Impul. | Defoc. | Glass | Motion | Zoom | Snow | Frost | Fog | Brit. | Contr. | Elast. | Pixel | JPEG | AVG |
|---|---|---|---|---|---|---|---|---|---|---|---|---|---|---|---|---|---|
| ResNet-50-GN | | 19.12 | 20.48 | 18.9 | 19.18 | 10.85 | 20.91 | 24.47 | 38.73 | 48.07 | 38.31 | 68.81 | 32.28 | 17.97 | 27.58 | 52.9 | 30.57 |
| VitBase | | 46.86 | 47.59 | 46.88 | 42.73 | 34.21 | 50.45 | 44.7 | 56.87 | 52.63 | 56.52 | 76.06 | 31.79 | 46.72 | 65.49 | 66.03 | 51.04 |
| Tent | | 23.42 | 33.3 | 36.89 | 20.69 | 18.35 | 26.8 | 33.22 | 39.82 | 47.14 | 47.95 | 68.8 | 43.37 | 26.82 | 46.03 | 57.00 | 37.97 |
| Tent | ✓ | 33.43 | 45.84 | 46.69 | 24.76 | 30.28 | 38.16 | 43.48 | 47.95 | 53.71 | 56.45 | 70.45 | 51.26 | 42.26 | 55.94 | 60.77 | 46.76 |
| SAR | | 19.00 | 21.33 | 21.29 | 19.94 | 12.57 | 23.02 | 25.69 | 39.99 | 43.7 | 41.36 | 68.79 | 37.34 | 18.76 | 36.46 | 54.05 | 32.22 |
| SAR | ✓ | 21.96 | 34.32 | 38.59 | 17.61 | 19.69 | 22.44 | 29.57 | 37.59 | 44.00 | 46.76 | 67.98 | 43.13 | 26.48 | 43.71 | 54.61 | 36.56 |
| COME | | 23.57 | 32.25 | 35.24 | 20.72 | 16.12 | 24.73 | 31.77 | 42.01 | 50.21 | 47.49 | 69.58 | 41.62 | 26.75 | 41.55 | 55.3 | 37.26 |
| COME | ✓ | 32.93 | 45.55 | 46.25 | 25.45 | 31.07 | 38.69 | 44.62 | 47.06 | 52.88 | 56.13 | 70.16 | 51.16 | 43.69 | 56.92 | 60.41 | 46.86 |
| ETA | | 25.34 | 38.44 | 40.96 | 21.61 | 21.48 | 29.35 | 37.24 | 40.48 | 48.5 | 49.11 | 67.52 | 46.94 | 30.88 | 48.79 | 57.88 | 40.30 |
| ETA | ✓ | 34.30 | 46.28 | 46.95 | 26.27 | 32.33 | 39.67 | 44.94 | 48.06 | 53.81 | 56.83 | 70.32 | 51.36 | 45.06 | 56.89 | 60.68 | 47.58 |
| CEMA | | 28.34 | 47.82 | 48.43 | 25.42 | 36.79 | 40.79 | 47.09 | 48.00 | 54.44 | 57.55 | 69.46 | 14.36 | 0.36 | 0.55 | 0.87 | 34.68 |
| CEMA | ✓ | 34.34 | 48.12 | 48.71 | 25.03 | 36.33 | 41.07 | 46.04 | 48.10 | 53.84 | 57.85 | 69.18 | 49.90 | 46.22 | 57.73 | 60.1 | 48.17 |

Table 5: Accuracy comparisons on the key components.

| Methods | Gauss. | Shot | Impul. | Defoc. | Glass | Motion | Zoom | Snow | Frost | Fog | Brit. | Contr. | Elastic | Pixel | JPEG | Avg. |
|---|---|---|---|---|---|---|---|---|---|---|---|---|---|---|---|---|
| Tent | 21.88 | 22.92 | 22.20 | 21.32 | 20.76 | 33.18 | 42.79 | 40.61 | 38.03 | 52.15 | 66.37 | 24.67 | 47.82 | 53.75 | 46.42 | 36.99 |
| AGR | 28.05 | 29.71 | 28.63 | 26.98 | 26.89 | 39.21 | 46.43 | 45.65 | 41.72 | 55.61 | 67.12 | 34.01 | 52.03 | 57.04 | 50.62 | 41.98 |
| UGR | 29.51 | 31.38 | 30.61 | 27.94 | 26.85 | 40.56 | 46.32 | 46.89 | 42.68 | 55.79 | 67.38 | 34.63 | 52.03 | 57.83 | 51.75 | 42.81 |
| AUGR | 34.14 | 36.16 | 35.45 | 32.16 | 32.43 | 44.44 | 49.13 | 49.58 | 44.77 | 57.85 | 66.83 | 39.97 | 54.65 | 59.04 | 53.67 | 46.02 |

the sparse-knowledge bottleneck and hinders robust knowledge accumulation. However, AUGR still improves Tent, SAR, COME, ETA, and CEMA by 8.79%, 4.34%, 9.60%, 7.28%, and 13.49%, respectively. Notably, AUGR remedies CEMA's failure cases on such "Elastic" (0.36%→46.22%), "Pixel"(0.55%→57.73%), and "JPEG"(0.87%→60.1%), indicating that it successfully mitigates the sparse-knowledge bottleneck and thereby achieves the best performance.

## 4.3 ABLATION STUDY AND ANALYSIS

All experiments in this subsection are performed on ImageNet-C (severity level 5).

**Contribution of key components.** To analyze the importance of each component, we evaluate the proposed agreement and uncertainty-guided reweighting (AUGR) in Table 5. Applying agreement-guided reweighting (AUG) or uncertainty-guided reweighting (UGR) alone already outperforms the baseline by 5.75% and 5.82%, indicating that they are beneficial to combat sparse knowledge bottleneck. The combination of them (AUGR) results in further improvements, supporting our design that emphasizes robust common knowledge while suppressing noisy uncommon ones. Moreover, we visualize the trends in mutual information for Tent and AUGR in Fig. 5, which reflect whether the model is accumulating useful knowledge. Tent is more vulnerable to noisy signals, making it harder to accumulate meaningful knowledge. In contrast, AUGR, with only about one-tenth of the updates (79 *vs* 728), nearly matches the knowledge accumulated by Tent using all data, further demonstrating its effectiveness in overcoming the sparse knowledge bottleneck.

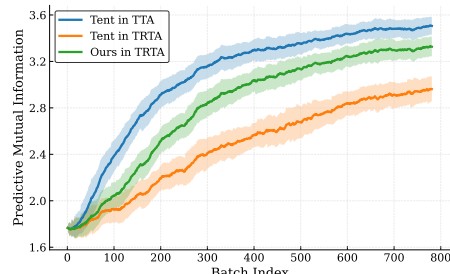

Figure 5: Comparison of mutual information between Tent and AUGR.

**Effect of adaptation delays.** In practice, communication further influence the adaptation delays. To emulate this constraint, we set equalize the adaptation delay across all methods, and summarize the results in Fig. 6. Baseline methods drop significantly with increasing adaptation latency because the sparse-knowledge bottleneck intensifies. Nevertheless, AUGR keeps them more robust even at extreme latency, improving Tent, ETA, and CEMA by 2.8%, 2.6%, and 3.4%, respectively. Detailed results are listed in Table 9 - 12.

**Effect of different model pairs.** In real-world applications, such as edge-cloud collaborative learning, small-capacity base models are favored for efficiency and flexibility, whereas high-capacity reference models are hosted in the cloud. To evaluate the applicability of AUGR, we further consider different model pairs with various capacities and present the results in Fig. 7. We observe consistent gains over all baselines, demonstrating that AUGR is architecture-agnostic. See Table 13 - 15 for details.

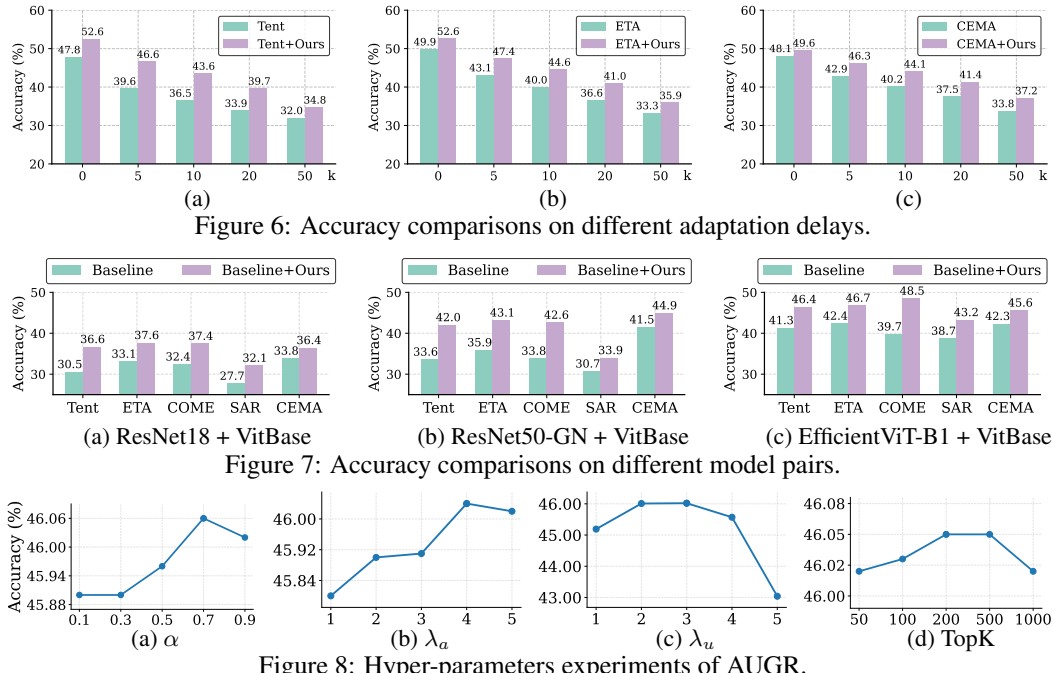

Figure 6: Accuracy comparisons on different adaptation delays.

(a) ResNet18 + VitBase  (b) ResNet50-GN + VitBase  (c) EfficientViT-B1 + VitBase

Figure 7: Accuracy comparisons on different model pairs.

(a) $\alpha$  (b) $\lambda_a$  (c) $\lambda_u$  (d) TopK

Figure 8: Hyper-parameters experiments of AUGR.

**Hyper-parameter Analysis.** We ablate the effect of hyperparameters in Fig. 8, including the coefficient $\alpha$ (Eq. 6), the TopK (Eq. 4), and the scale factors $\lambda_a$ and $\lambda_u$ (Eq. 9). Performance generally increases as these values grow, while excessively large velues can degrade the accuracy slightly. From Figs. 8 (b) and (c), we notice that using a large scale and setting $\lambda_a > \lambda_u$ yields better performance, highlighting the importance of the agreement-guided reweighting. Overall, our method is robust to hyper-parameters.

**Comparison with training-free approaches.** Training-free approaches achieve competitive performance while maintaining high efficiency. (Iwasawa & Matsuo, 2021; Boudiaf et al., 2022; Niu et al., 2024). We report the mean classification accuracy on full ImageNet-C and the average wall-clock time to process 50,000 samples (with Gaussian Noise) in Fig. 9. Experiments are conducted on a single ViTBase following (Niu et al., 2024). We find that

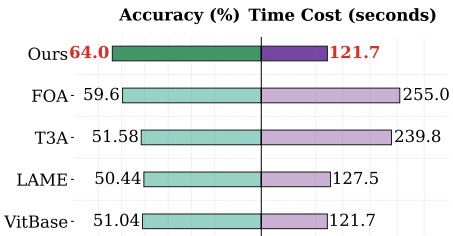

Figure 9: Ours *vs* training-free approaches.

training-free approaches do not necessarily reduce end-to-end latency. Concretely, T3A incurs substantial overhead for per-class support-set selection, so its cost scales with the number of classes; FOA's runtime scales with the number of forward passes, and more passes directly increase latency. In contrast, our method attains superior accuracy while maintaining efficiency, suggesting that TRTA is a realistic paradigm for TTA.

## 5 CONCLUSION

In this paper, we propose a realistic TTA paradigm, Test-Real-Time Adaptation (TRTA), in which prediction on the current input and model adaptation proceed simultaneously without synchronization pauses. TRTA inherently entails sparse adaptation, leaving few opportunities to correct earlier errors and thereby triggering a sparse-knowledge bottleneck. To mitigate this challenge, we propose Agreement- and Uncertainty-Guided Reweighting (AUGR), which integrates AGR and UGR as complementary signals to produce reliable self-training supervision, thereby enabling existing TTA methods to operate effectively under TRTA. AGR measures inter-model agreement via the concordance of predicted class rankings between the base and reference models, highlighting knowledge consistently supported by both; UGR then calibrates this signal by down-weighting low-confidence predictions. Extensive experiments on ImageNet-C/R/K validate the effectiveness of AUGR in addressing the sparse-knowledge bottleneck under TRTA.

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

# Appendix

## A  THE USE OF LARGE LANGUAGE MODELS (LLMs)

LLMs were used exclusively for language polishing, such as grammar, clarity, and readability. They were not used for ideation, methodology, data handling, analysis, code, or results. This manuscript was reviewed and approved by the authors, who take full responsibility for the content. LLMs are not authors.

## B  BENCHMARK DETAILS

**ImageNet-C**. ImageNet-C (Hendrycks & Dietterich, 2019) augments the ImageNet validation set with 15 common corruption types, including Gaussian, Shot, and Impulse noise, Defocus, Glass, Motion, Zoom blur, Snow, Frost, Fog, Brightness, Contrast, Elastic transform, Pixelate, and JPEG compression. Each corruption is processed at five severity levels, yielding 50,000 images per type and 750,000 images in total across 1,000 classes.

**ImageNet-Rendition (R)**. ImageNet-R (Hendrycks et al., 2021) contains artistic renditions of ImageNet categories, including cartoons, paintings, origami, embroidery, toys, and sculptures, covering 200 classes with 30,000 images.

**ImageNet-Sketch (K)**. ImageNet-K (Wang et al., 2019) provides hand-drawn sketch depictions of objects from the ImageNet label space. The dataset comprises 50,000 sketches spanning the 1,000 ImageNet categories.

## C  IMPLEMENTATION DETAILS

**TENT.** TENT (Wang et al., 2021) adapts a model by minimizing the entropy of target samples. Implementation follows the official code.

**ETA.** ETA (Niu et al., 2022) selects a subset of incoming target data and performs sample-wise weighted entropy minimization. The exponential moving average (EMA) factor, cosine-similarity threshold, and entropy threshold are set to $0.9$, $0.05$, and $0.4 \times \log(C)$, respectively, where $C$ is the number of classes. Implementation follows the official code.

**SAR.** SAR (Niu et al., 2023) improves stability by using batch-agnostic normalization (group/layer norm) and a sharpness-aware reliable entropy loss: it filters noisy samples with large gradients and encourages the weights to converge to a flatter minimum. The same entropy threshold is identical to ETA. Trainable parameters are the affine parameters of layer normalization in blocks 1–8 for ViT-Base, and the batch/group normalization layers in layer 1–3 for ResNet-50. Implementation follows the official code.

**COME.** COME (Zhang et al., 2025) regularizes confidence to be conservative on unreliable samples by recovering logits under an $\ell_p$ constraint. COME has two hyper-parameters: $p$ (the $p$-norm) and $\tau$ (the magnitude of recovered logits), which we set to $p = 2$ and $\tau = 1$. Implementation follows the official code.

**CEMA.** CEMA (Chen et al., 2024c) is an edge–cloud TTA baseline: the edge model actively selects a small set of online samples to upload to a cloud buffer; the cloud model is optimized by entropy minimization, and a copy of the edge model is updated using entropy minimization plus a KL term. We set the EMA factor to $0.9$, the cosine-similarity threshold to $0.05$, and the entropy threshold to $0.4 \times \log(C)$. The remaining hyper-parameters are $\alpha = 3$, $\beta = 3$, $\gamma = 1$, temperature$= 1$, and buffer size$= 2000$. Implementation follows the official code.

**LAME.** LAME (Boudiaf et al., 2022) adopts adjusts the output logits using an efficient concave-convex procedure. For the hyper-parameters: $k$ in the kNN affinity matrix is set to 5. Implementation follows the official code.

**T3A.** T3A (Iwasawa & Matsuo, 2021) maintains a memory-based classifier from selected, $l_2$-normalized features and, after each update, adjusts the output logits with this classifier. The size of

support set for each class is chosen from $\{1, 5, 20, 50, 100\}$. We fix it to 20 for the optimal results. Implementation follows the official code.

**FOA.** FOA (Niu et al., 2024) designs an activation shifting scheme that directly tunes the model activations for shifted target-domain data, making it align with the source domain. The number of forward $k$ is set to 2. Implementation follows the official code.

**AUGR.** To evaluate AUGR in isolation, Tent's original loss is omitted and Eq. (10) is applied alone. Moreover, AUGR can be seamlessly plugged into existing TTA methods by appending its loss term to the baseline objectives. For the experiments in Table 1, TopK in Eq. 4, the coefficient $\alpha$ in Eq. 7, the entropy margin m in Eq. 8, the scale factors $\lambda_a$ and $\lambda_u$ in Eq. 9 are 50, 0.9, $0.4 \times \log(C)$, 5, and 3, respectively. When integrated with baseline methods, $\lambda_a$ and $\lambda_u$ are reduced to 2 and 1.

**Pre-trained model.** In line with previous works (Niu et al., 2022; 2023), we use ResNet-50 with batch/group normalization and vision transformers as the architecture of the pre-trained models for most cases. To prove the applicability of our method, we additionally test it on ResNet-18 with batch normalization and EfficientVit (Cai et al., 2022). Specifically, ResNet-18-BN and ResNet-50-BN are obtained from `Pytorch` library; ResNet-50-GN , VitBase and EfficientVit are obtained from `timm` repository.

**Optimizer.** We use SGD optimizer with a momentum of 0.9 for the base and reference models. For ResNet-50 and VitBase, the learning rates are 0.00025 and 0.001, respectively. The batch size is 64.

## C.1 MORE EXPERIMENTAL RESULTS

**Results on VitBase.** In this case, we use a historical base-model checkpoint as the reference. Results appear in Table 6.

**Detailed results.** This subsection contains detailed results for figures or tables in the main manuscript. Tables 9, 10, 11, 12 correspond to Fig. 6. Tables 13, 14, 15 correspond to Figure 7.

Table 6: Accuracy comparisons in TRTA on ImageNet-C with a single VitBase.

| Methods | XXX | Gauss. | Shot | Impul. | Defoc. | Glass | Motion | Zoom | Snow | Frost | Fog | Brit. | Contr. | Elast. | Pixel | JPEG | AVG | #Updates |
|---|---|---|---|---|---|---|---|---|---|---|---|---|---|---|---|---|---|---|
| VitBase | ✗ | 46.86 | 47.59 | 46.88 | 42.73 | 34.21 | 50.45 | 44.70 | 56.87 | 52.63 | 56.52 | 76.06 | 31.79 | 46.72 | 65.49 | 66.03 | 51.04 | 0 |
| Tent | ✗ | 55.08 | 55.75 | 55.68 | 52.82 | 46.63 | 56.63 | 52.45 | 62.41 | 57.59 | 62.87 | 77.43 | 56.82 | 55.23 | 69.24 | 67.54 | 58.94 | 196 |
| | ✓ | 57.91 | 58.98 | 58.91 | 57.33 | 55.48 | 61.52 | 58.35 | 67.29 | 65.58 | 69.45 | 79.44 | 60.16 | 64.44 | 73.60 | 71.53 | 64.00 | 156 |
| | △ | 2.83↑ | 3.23↑ | 3.23↑ | 4.51↑ | 8.85↑ | 4.89↑ | 5.90↑ | 4.88↑ | 7.99↑ | 6.58↑ | 2.01↑ | 3.34↑ | 9.21↑ | 4.36↑ | 3.99↑ | 5.06↑ | 40↓ |
| SAR | ✗ | 53.49 | 54.63 | 54.27 | 51.53 | 44.92 | 55.34 | 50.81 | 61.10 | 57.82 | 60.74 | 77.02 | 48.37 | 52.99 | 68.36 | 66.94 | 57.22 | 131 |
| | ✓ | 57.63 | 59.07 | 58.61 | 56.54 | 53.73 | 61.02 | 57.66 | 66.34 | 64.63 | 67.39 | 79.07 | 56.52 | 62.81 | 73.04 | 70.86 | 62.99 | 112 |
| | △ | 4.14↑ | 4.44↑ | 4.34↑ | 5.01↑ | 8.81↑ | 5.68↑ | 6.85↑ | 5.24↑ | 6.81↑ | 6.65↑ | 2.05↑ | 8.15↑ | 9.82↑ | 4.68↑ | 3.92↑ | 5.77↑ | 19↓ |
| COME | ✗ | 56.64 | 57.84 | 57.80 | 56.82 | 54.94 | 61.26 | 58.33 | 65.99 | 65.00 | 69.47 | 79.05 | 53.90 | 64.76 | 72.89 | 70.74 | 63.03 | 196 |
| | ✓ | 58.39 | 59.77 | 59.21 | 58.05 | 57.28 | 63.01 | 60.32 | 68.30 | 66.44 | 70.58 | 79.45 | 60.72 | 66.88 | 74.29 | 72.11 | 64.99 | 156 |
| | △ | 1.75↑ | 1.93↑ | 1.41↑ | 1.23↑ | 2.34↑ | 1.75↑ | 1.99↑ | 2.31↑ | 1.44↑ | 1.11↑ | 0.40↑ | 6.82↑ | 2.12↑ | 1.40↑ | 1.37↑ | 1.96↑ | 40↓ |
| ETA | ✗ | 58.12 | 58.83 | 59.11 | 57.64 | 55.27 | 61.52 | 58.94 | 66.84 | 65.22 | 68.75 | 79.12 | 61.30 | 65.11 | 73.03 | 71.03 | 63.99 | 196 |
| | ✓ | 58.94 | 60.16 | 59.77 | 58.38 | 57.33 | 62.91 | 60.42 | 68.25 | 66.67 | 70.92 | 79.49 | 61.29 | 67.85 | 74.23 | 72.38 | 65.27 | 156 |
| | △ | 0.82↑ | 1.33↑ | 0.66↑ | 0.74↑ | 2.06↑ | 1.39↑ | 1.48↑ | 1.41↑ | 1.45↑ | 2.17↑ | 0.37↑ | 0.01↓ | 2.74↑ | 1.20↑ | 1.35↑ | 1.28↑ | 40↓ |

Table 7: Accuracy comparisons in TRTA. A historical base model serves as the reference model.

| Methods | AUGR | Gauss. | Shot | Impul. | Defoc. | Glass | Motion | Zoom | Snow | Frost | Fog | Brit. | Contr. | Elast. | Pixel | JPEG | AVG | #Updates |
|---|---|---|---|---|---|---|---|---|---|---|---|---|---|---|---|---|---|---|
| ResNet-50 | ✗ | 2.21 | 2.93 | 1.85 | 17.92 | 9.82 | 14.78 | 22.49 | 16.89 | 23.30 | 24.43 | 58.93 | 5.43 | 16.95 | 20.60 | 31.64 | 18.01 | 0 |
| Tent | ✗ | 22.12 | 23.39 | 22.88 | 21.58 | 21.37 | 34.12 | 45.27 | 41.29 | 38.06 | 54.17 | 66.35 | 24.83 | 50.02 | 54.88 | 47.60 | 37.86 | 196 |
| | ✓ | 27.13 | 28.97 | 27.82 | 25.76 | 25.93 | 40.22 | 48.92 | 46.89 | 41.61 | 57.65 | 67.20 | 27.07 | 54.60 | 58.46 | 51.87 | 42.01 | 156 |
| | △ | 5.01↑ | 5.58↑ | 4.94↑ | 4.18↑ | 4.56↑ | 6.10↑ | 3.65↑ | 5.60↑ | 3.55↑ | 3.48↑ | 0.85↑ | 2.24↑ | 4.58↑ | 3.58↑ | 4.27↑ | 4.15↑ | 40↓ |
| SAR | ✗ | 19.24 | 20.89 | 19.67 | 19.09 | 19.24 | 31.35 | 42.76 | 38.36 | 36.35 | 51.76 | 65.87 | 22.54 | 47.28 | 52.39 | 44.66 | 35.43 | 98 |
| | ✓ | 24.54 | 26.88 | 25.90 | 24.54 | 24.32 | 37.77 | 47.61 | 44.24 | 40.34 | 56.43 | 67.16 | 30.19 | 52.59 | 57.20 | 50.29 | 40.67 | 87 |
| | △ | 5.30↑ | 5.99↑ | 6.23↑ | 5.45↑ | 5.08↑ | 6.42↑ | 4.85↑ | 5.88↑ | 3.99↑ | 4.67↑ | 1.29↑ | 7.65↑ | 5.31↑ | 4.81↑ | 5.63↑ | 5.24↑ | 11↓ |
| COME | ✗ | 25.36 | 27.42 | 27.54 | 24.56 | 26.06 | 40.17 | 48.48 | 46.04 | 41.76 | 56.60 | 67.09 | 36.48 | 53.41 | 57.46 | 51.14 | 41.97 | 196 |
| | ✓ | 29.40 | 31.51 | 29.97 | 26.92 | 27.15 | 42.52 | 50.01 | 48.30 | 43.03 | 58.16 | 67.32 | 38.49 | 55.37 | 59.18 | 52.84 | 44.01 | 156 |
| | △ | 4.04↑ | 4.09↑ | 2.43↑ | 2.36↑ | 1.09↑ | 2.35↑ | 1.53↑ | 2.26↑ | 1.27↑ | 1.56↑ | 0.23↑ | 2.01↑ | 1.96↑ | 1.72↑ | 1.70↑ | 2.04↑ | 40↓ |
| ETA | ✗ | 27.28 | 31.22 | 29.98 | 27.61 | 28.13 | 41.29 | 49.64 | 47.17 | 42.64 | 57.46 | 67.39 | 37.89 | 54.48 | 58.41 | 52.09 | 43.51 | 196 |
| | ✓ | 30.22 | 32.13 | 29.91 | 27.51 | 27.66 | 42.92 | 49.83 | 48.33 | 42.85 | 58.22 | 67.04 | 37.90 | 55.38 | 58.98 | 52.72 | 44.11 | 156 |
| | △ | 2.94↑ | 0.91↑ | 0.07↓ | 0.10↓ | 0.47↓ | 1.63↑ | 0.19↑ | 1.16↑ | 0.21↑ | 0.76↑ | 0.35↓ | 0.01↑ | 0.90↑ | 0.57↑ | 0.63↑ | 0.60↑ | 40↓ |

Table 8: Accuracy comparisons in TRTA. A frozen VitBase serves as the reference model.

| Methods | AUGR | Gauss. | Shot | Impul. | Defoc. | Glass | Motion | Zoom | Snow | Frost | Fog | Brit. | Contr. | Elast. | Pixel | JPEG | AVG | #Updates |
|---|---|---|---|---|---|---|---|---|---|---|---|---|---|---|---|---|---|---|
| Tent | ✗ | 24.11 | 25.62 | 24.23 | 22.70 | 21.79 | 34.96 | 43.81 | 42.40 | 38.96 | 52.99 | 66.62 | 26.22 | 48.54 | 54.80 | 48.03 | 38.38 | 112 |
| | ✓ | 34.05 | 35.55 | 34.54 | 30.89 | 30.21 | 43.82 | 48.13 | 49.56 | 44.21 | 57.04 | 67.72 | 36.79 | 52.94 | 59.26 | 53.68 | 45.23 | 112 |
| | △ | 9.94↑ | 9.93↑ | 10.31↑ | 8.19↑ | 8.42↑ | 8.86↑ | 4.32↑ | 7.16↑ | 5.25↑ | 4.05↑ | 1.10↑ | 10.57↑ | 4.40↑ | 4.46↑ | 5.65↑ | 6.85↑ | 0 |
| SAR | ✗ | 17.81 | 19.63 | 19.37 | 17.37 | 18.81 | 29.91 | 41.84 | 37.71 | 36.09 | 50.42 | 66.13 | 17.44 | 46.51 | 51.68 | 43.65 | 34.29 | 72 |
| | ✓ | 28.47 | 31.07 | 29.89 | 24.37 | 26.62 | 39.19 | 47.36 | 46.35 | 41.38 | 55.92 | 67.05 | 20.05 | 52.84 | 57.52 | 50.95 | 41.27 | 72 |
| | △ | 10.66↑ | 11.44↑ | 10.52↑ | 7.00↑ | 7.81↑ | 9.28↑ | 5.52↑ | 8.64↑ | 5.29↑ | 5.50↑ | 0.92↑ | 2.61↑ | 6.33↑ | 5.84↑ | 7.30↑ | 6.98↑ | 0 |
| COME | ✗ | 26.46 | 28.62 | 26.60 | 23.74 | 23.85 | 38.49 | 46.58 | 45.38 | 41.15 | 55.20 | 67.09 | 30.78 | 51.25 | 56.69 | 50.38 | 40.82 | 112 |
| | ✓ | 33.16 | 34.69 | 33.83 | 29.76 | 29.35 | 43.36 | 47.59 | 48.84 | 43.68 | 56.45 | 67.56 | 35.77 | 51.99 | 59.14 | 53.57 | 44.58 | 112 |
| | △ | 6.70↑ | 6.07↑ | 7.23↑ | 6.02↑ | 5.50↑ | 4.87↑ | 1.01↑ | 3.46↑ | 2.53↑ | 1.25↑ | 0.47↑ | 4.99↑ | 0.74↑ | 2.45↑ | 3.19↑ | 3.76↑ | 0 |
| ETA | ✗ | 28.02 | 30.12 | 28.11 | 25.49 | 25.98 | 39.70 | 47.42 | 46.32 | 41.91 | 55.99 | 67.24 | 32.21 | 52.45 | 57.21 | 50.95 | 41.94 | 112 |
| | ✓ | 33.66 | 35.22 | 34.08 | 30.40 | 30.23 | 43.47 | 48.01 | 49.26 | 43.94 | 56.83 | 67.69 | 36.23 | 52.50 | 59.21 | 53.57 | 44.95 | 112 |
| | △ | 5.64↑ | 5.10↑ | 5.97↑ | 4.91↑ | 4.25↑ | 3.77↑ | 0.59↑ | 2.94↑ | 2.03↑ | 0.84↑ | 0.45↑ | 4.02↑ | 0.05↑ | 2.00↑ | 2.62↑ | 3.01↑ | 0 |
| CEMA | ✗ | 25.29 | 34.37 | 33.19 | 20.84 | 25.62 | 38.20 | 46.84 | 43.40 | 40.92 | 51.40 | 65.04 | 18.32 | 48.76 | 54.91 | 50.62 | 39.85 | 71 |
| | ✓ | 30.90 | 36.37 | 34.46 | 27.29 | 28.19 | 40.74 | 47.81 | 46.40 | 42.92 | 54.23 | 65.43 | 30.09 | 50.89 | 56.83 | 52.37 | 42.99 | 56 |
| | △ | 5.61↑ | 2.00↑ | 1.27↑ | 6.45↑ | 2.57↑ | 2.54↑ | 0.97↑ | 3.00↑ | 2.00↑ | 2.83↑ | 0.39↑ | 11.77↑ | 2.13↑ | 1.92↑ | 1.75↑ | 3.14↑ | 15↓ |

Table 9: Accuracy comparisons in TRTA on ImageNet-C with different adaptation delays ($k$=5).

| Methods | AUGR | Gauss. | Shot | Impul. | Defoc. | Glass | Motion | Zoom | Snow | Frost | Fog | Brit. | Contr. | Elast. | Pixel | JPEG | AVG |
|---|---|---|---|---|---|---|---|---|---|---|---|---|---|---|---|---|---|
| ResNet-50 | ✗ | 2.21 | 2.93 | 1.85 | 17.92 | 9.82 | 14.78 | 22.49 | 16.89 | 23.30 | 24.43 | 58.93 | 5.43 | 16.95 | 20.60 | 31.64 | 18.01 |
| VitBase | ✗ | 46.86 | 47.59 | 46.88 | 42.73 | 34.21 | 50.45 | 44.70 | 56.87 | 52.63 | 56.52 | 76.06 | 31.79 | 46.72 | 65.49 | 66.03 | 51.04 |
| Tent | ✗ | 25.05 | 26.58 | 25.63 | 24.30 | 23.37 | 36.14 | 44.84 | 43.32 | 39.77 | 53.66 | 66.73 | 30.79 | 49.85 | 55.44 | 48.62 | 39.61 |
| | ✓ | 34.40 | 36.57 | 35.79 | 33.21 | 32.10 | 45.09 | 49.65 | 50.26 | 45.27 | 58.07 | 67.85 | 42.50 | 54.96 | 59.79 | 54.09 | 46.64 |
| ETA | ✗ | 29.03 | 30.91 | 30.50 | 27.70 | 27.80 | 40.37 | 48.11 | 47.09 | 42.52 | 56.64 | 67.34 | 36.12 | 53.72 | 57.81 | 51.57 | 43.15 |
| | ✓ | 35.08 | 37.34 | 36.69 | 34.03 | 33.22 | 46.00 | 50.99 | 50.95 | 45.77 | 58.73 | 67.89 | 43.74 | 55.99 | 60.03 | 54.55 | 47.40 |
| CEMA | ✗ | 27.87 | 37.57 | 35.86 | 23.75 | 31.22 | 42.09 | 48.88 | 46.06 | 43.19 | 54.35 | 65.24 | 24.27 | 53.09 | 57.10 | 52.45 | 42.87 |
| | ✓ | 33.77 | 39.19 | 38.08 | 31.99 | 34.10 | 44.62 | 49.89 | 48.70 | 44.93 | 56.42 | 65.62 | 39.91 | 54.63 | 58.36 | 53.75 | 46.26 |

Table 10: Accuracy comparisons in TRTA on ImageNet-C with different adaptation delays ($k$=10).

| Methods | AUGR | Gauss. | Shot | Impul. | Defoc. | Glass | Motion | Zoom | Snow | Frost | Fog | Brit. | Contr. | Elast. | Pixel | JPEG | AVG |
|---|---|---|---|---|---|---|---|---|---|---|---|---|---|---|---|---|---|
| Tent | | 21.33 | 22.37 | 21.44 | 20.53 | 20.07 | 32.67 | 42.53 | 40.31 | 37.46 | 51.83 | 66.39 | 24.28 | 47.49 | 53.42 | 45.95 | 36.54 |
| | ✓ | 30.83 | 32.34 | 31.03 | 28.54 | 27.77 | 41.18 | 47.64 | 47.82 | 42.96 | 56.16 | 67.70 | 37.06 | 52.36 | 58.04 | 52.16 | 43.57 |
| ETA | | 25.11 | 27.35 | 26.19 | 22.09 | 24.79 | 37.13 | 46.03 | 44.22 | 40.47 | 54.63 | 66.95 | 28.61 | 51.17 | 56.07 | 49.15 | 40.00 |
| | ✓ | 31.68 | 33.57 | 32.40 | 29.50 | 29.95 | 42.65 | 48.96 | 48.87 | 43.63 | 56.80 | 67.75 | 37.99 | 53.92 | 58.65 | 52.69 | 44.60 |
| CEMA | | 24.51 | 34.80 | 33.69 | 21.71 | 26.71 | 38.10 | 47.86 | 42.97 | 41.18 | 52.66 | 64.39 | 18.28 | 49.97 | 55.33 | 50.89 | 40.20 |
| | ✓ | 30.79 | 37.63 | 35.55 | 28.12 | 30.89 | 42.03 | 49.03 | 47.25 | 42.77 | 55.06 | 65.19 | 35.42 | 52.34 | 57.20 | 52.55 | 44.12 |

Table 11: Accuracy comparisons in TRTA on ImageNet-C with different adaptation delays ($k$=20).

| Methods | AUGR | Gauss. | Shot | Impul. | Defoc. | Glass | Motion | Zoom | Snow | Frost | Fog | Brit. | Contr. | Elast. | Pixel | JPEG | AVG |
|---|---|---|---|---|---|---|---|---|---|---|---|---|---|---|---|---|---|
| Tent | | 18.32 | 19.03 | 18.58 | 17.58 | 17.45 | 29.14 | 40.83 | 37.46 | 35.52 | 49.71 | 66.01 | 18.41 | 45.57 | 51.50 | 43.28 | 33.89 |
| | ✓ | 25.79 | 27.04 | 25.78 | 23.93 | 22.96 | 36.28 | 44.65 | 44.21 | 40.17 | 53.88 | 66.93 | 28.89 | 49.48 | 55.93 | 49.00 | 39.66 |
| ETA | | 22.05 | 22.74 | 20.81 | 19.20 | 20.41 | 32.44 | 43.18 | 40.75 | 37.78 | 52.43 | 66.46 | 22.69 | 48.10 | 53.73 | 46.11 | 36.59 |
| | ✓ | 27.31 | 28.68 | 27.22 | 25.06 | 24.91 | 37.90 | 46.08 | 45.39 | 41.28 | 54.83 | 67.11 | 31.64 | 51.06 | 56.58 | 49.96 | 41.00 |
| CEMA | | 21.78 | 30.67 | 30.20 | 19.59 | 23.81 | 31.11 | 42.63 | 40.56 | 39.20 | 49.94 | 64.64 | 18.23 | 48.61 | 53.97 | 48.09 | 37.53 |
| | ✓ | 27.05 | 33.63 | 33.15 | 25.13 | 27.16 | 36.74 | 46.62 | 44.53 | 41.12 | 52.53 | 65.21 | 29.88 | 50.90 | 56.05 | 50.75 | 41.36 |

Table 12: Accuracy comparisons in TRTA on ImageNet-C with different adaptation delays ($k$=50).

| Methods | AUGR | Gauss. | Shot | Impul. | Defoc. | Glass | Motion | Zoom | Snow | Frost | Fog | Brit. | Contr. | Elast. | Pixel | JPEG | AVG |
|---|---|---|---|---|---|---|---|---|---|---|---|---|---|---|---|---|---|
| Tent | | 16.25 | 16.83 | 16.66 | 15.57 | 15.94 | 27.09 | 39.38 | 35.28 | 34.19 | 48.16 | 65.80 | 14.08 | 44.37 | 49.67 | 40.93 | 32.01 |
| | ✓ | 19.45 | 20.33 | 19.44 | 18.19 | 17.76 | 30.63 | 41.38 | 38.88 | 36.30 | 50.54 | 66.19 | 18.43 | 46.35 | 52.78 | 44.73 | 34.76 |
| ETA | | 17.41 | 18.50 | 18.02 | 16.21 | 16.77 | 28.51 | 40.73 | 36.50 | 35.18 | 49.77 | 65.99 | 16.57 | 45.76 | 50.66 | 42.54 | 33.28 |
| | ✓ | 20.44 | 21.97 | 20.94 | 19.00 | 18.94 | 31.99 | 42.39 | 39.95 | 37.29 | 51.58 | 66.35 | 21.15 | 47.72 | 53.49 | 45.63 | 35.92 |
| CEMA | | 16.84 | 17.74 | 17.61 | 17.47 | 19.91 | 29.66 | 40.86 | 38.47 | 37.75 | 47.17 | 64.72 | 17.17 | 46.95 | 51.28 | 43.75 | 33.82 |
| | ✓ | 22.04 | 22.91 | 26.84 | 21.51 | 22.88 | 32.82 | 41.90 | 40.55 | 38.87 | 49.60 | 65.10 | 22.81 | 48.40 | 53.96 | 47.27 | 37.16 |

## C.2 MORE ABLATION STUDIES.

**The choice of reference model.** To analyze the effect of the reference model, we consider two additional ablations: single-model adaptation and collaborative adaptation with a frozen foundation model, reported in Table. 7 and Table 8, respectively. In the first case, we use a historical version of the base model as the reference to provide agreement assessment, which aligns with our design principle of highlighting robust common knowledge. The reference requires no backward computation, but obtaining its logits incurs an extra forward pass and thus reduces the total number of updates. Nevertheless, as shown in Table 7, our method still consistently improves existing approaches across most domain shifts.

In the second case, the overall updates increase compared to Table 1, since the frozen reference model saves one backward pass. As shown in Table 8, additional updates provide marginal improvements for the baseline methods, suggesting they accumulate knowledge ineffectively. In contrast, our method significantly enhances their robustness under diverse domain shifts. The above empirical results demonstrate that our method does not rely on a specific reference model and highlight the effectiveness of our design, which promotes robust common knowledge to combat the sparse knowledge bottleneck.

Table 13: Accuracy comparisons in TRTA on ImageNet-C with ResNet-18&VitBase.

| Methods | AUGR | Gauss. | Shot | Impul. | Defoc. | Glass | Motion | Zoom | Snow | Frost | Fog | Brit. | Contr. | Elast. | Pixel | JPEG | AVG |
|---|---|---|---|---|---|---|---|---|---|---|---|---|---|---|---|---|---|
| ResNet-18 | | 1.16 | 1.80 | 1.00 | 11.44 | 8.68 | 11.16 | 17.65 | 10.87 | 16.45 | 14.29 | 51.30 | 3.44 | 16.78 | 3.12 | 29.64 | 14.59 |
| VitBase | | 46.86 | 47.59 | 46.88 | 42.73 | 34.21 | 50.45 | 44.7 | 56.87 | 52.63 | 56.52 | 76.06 | 31.79 | 46.72 | 65.49 | 66.03 | 51.04 |
| Tent | ✓ | 17.11 | 18.29 | 17.14 | 14.99 | 16.14 | 26.84 | 35.75 | 32.93 | 30.99 | 43.53 | 58.58 | 17.04 | 40.52 | 46.72 | 41.25 | 30.52 |
| | | 25.23 | 27.14 | 24.95 | 22.04 | 22.92 | 33.29 | 39.74 | 39.14 | 35.91 | 48.65 | 59.56 | 27.91 | 45.53 | 51.21 | 46.33 | 36.64 |
| SAR | ✓ | 13.84 | 15.74 | 14.36 | 12.06 | 13.33 | 23.74 | 33.90 | 30.11 | 29.03 | 40.77 | 58.29 | 9.56 | 38.59 | 43.97 | 37.74 | 27.67 |
| | | 18.35 | 22.02 | 19.31 | 15.83 | 17.15 | 28.72 | 37.79 | 34.79 | 32.32 | 45.66 | 58.97 | 16.04 | 42.84 | 48.36 | 42.68 | 32.06 |
| COME | ✓ | 19.36 | 21.09 | 19.76 | 16.56 | 17.90 | 28.75 | 37.44 | 35.09 | 32.78 | 45.36 | 58.86 | 18.35 | 42.65 | 48.57 | 43.07 | 32.37 |
| | | 26.21 | 27.88 | 26.18 | 22.81 | 24.07 | 34.43 | 40.67 | 39.75 | 36.50 | 49.17 | 59.82 | 28.99 | 46.63 | 51.64 | 46.76 | 37.43 |
| ETA | ✓ | 20.60 | 22.28 | 20.74 | 16.31 | 18.96 | 29.61 | 38.15 | 35.73 | 33.37 | 46.14 | 59.10 | 19.07 | 43.32 | 48.98 | 43.60 | 33.06 |
| | | 26.47 | 28.27 | 26.41 | 23.14 | 24.43 | 34.34 | 40.86 | 39.90 | 36.78 | 49.31 | 59.69 | 29.40 | 46.77 | 51.65 | 46.84 | 37.62 |
| CEMA | ✓ | 22.01 | 29.70 | 28.50 | 17.84 | 20.03 | 30.89 | 39.23 | 35.39 | 34.55 | 44.15 | 57.32 | 11.18 | 43.23 | 48.48 | 44.66 | 33.81 |
| | | 25.21 | 31.11 | 29.58 | 21.27 | 23.24 | 33.44 | 40.70 | 38.01 | 35.87 | 45.72 | 57.57 | 22.91 | 44.70 | 50.29 | 46.31 | 36.39 |

Table 14: Accuracy comparisons in TRTA on ImageNet-C with ResNet50-GN&VitBase.

| Methods | AUGR | Gauss. | Shot | Impul. | Defoc. | Glass | Motion | Zoom | Snow | Frost | Fog | Brit. | Contr. | Elast. | Pixel | JPEG | AVG |
|---|---|---|---|---|---|---|---|---|---|---|---|---|---|---|---|---|---|
| ResNet-50-GN | | 19.12 | 20.48 | 18.9 | 19.18 | 10.85 | 20.91 | 24.47 | 38.73 | 48.07 | 38.31 | 68.81 | 32.28 | 17.97 | 27.58 | 52.9 | 30.57 |
| VitBase | | 46.86 | 47.59 | 46.88 | 42.73 | 34.21 | 50.45 | 44.7 | 56.87 | 52.63 | 56.52 | 76.06 | 31.79 | 46.72 | 65.49 | 66.03 | 51.04 |
| Tent | ✓ | 23.35 | 25.25 | 24.05 | 19.97 | 14.02 | 24.58 | 27.04 | 40.39 | 45.82 | 37.65 | 69.19 | 36.89 | 22.06 | 39.45 | 53.73 | 33.56 |
| | | 33.76 | 35.71 | 34.95 | 26.56 | 22.72 | 33.48 | 34.79 | 48.04 | 49.44 | 51.32 | 70.89 | 44.42 | 35.90 | 51.82 | 55.89 | 41.98 |
| SAR | ✓ | 19.00 | 20.90 | 19.34 | 18.96 | 11.30 | 21.54 | 24.86 | 38.24 | 46.38 | 36.40 | 68.80 | 33.60 | 18.08 | 29.34 | 53.10 | 30.66 |
| | | 22.98 | 27.49 | 25.71 | 17.62 | 14.82 | 25.88 | 27.78 | 39.67 | 44.03 | 37.34 | 69.68 | 37.24 | 22.06 | 42.62 | 54.24 | 33.94 |
| COME | ✓ | 23.68 | 25.16 | 24.36 | 20.53 | 13.41 | 23.94 | 26.61 | 41.14 | 48.74 | 42.54 | 69.05 | 35.98 | 22.20 | 35.85 | 53.50 | 33.78 |
| | | 33.99 | 37.04 | 35.58 | 25.91 | 24.61 | 34.98 | 36.57 | 47.63 | 47.36 | 48.25 | 71.31 | 46.44 | 38.17 | 54.40 | 56.47 | 42.58 |
| ETA | ✓ | 25.73 | 28.68 | 26.10 | 22.13 | 17.38 | 27.83 | 29.11 | 42.32 | 44.63 | 40.22 | 69.57 | 39.67 | 27.42 | 42.71 | 54.28 | 35.85 |
| | | 34.93 | 37.02 | 36.06 | 28.16 | 25.11 | 35.03 | 36.35 | 49.00 | 49.91 | 49.74 | 71.16 | 46.06 | 38.64 | 53.21 | 56.52 | 43.13 |
| CEMA | ✓ | 30.94 | 39.35 | 38.32 | 22.90 | 18.62 | 32.33 | 37.59 | 46.63 | 52.04 | 49.36 | 71.94 | 45.86 | 30.64 | 46.26 | 59.00 | 41.45 |
| | | 36.97 | 41.52 | 40.79 | 26.85 | 24.01 | 36.62 | 40.03 | 49.52 | 51.77 | 53.42 | 72.33 | 46.56 | 39.88 | 54.12 | 59.01 | 44.89 |

Table 15: Accuracy comparisons in TRTA on ImageNet-C with EfficientViT-B1&VitBase.

| Methods | AUGR | Gauss. | Shot | Impul. | Defoc. | Glass | Motion | Zoom | Snow | Frost | Fog | Brit. | Contr. | Elast. | Pixel | JPEG | AVG |
|---|---|---|---|---|---|---|---|---|---|---|---|---|---|---|---|---|---|
| EfficientViT-B1 | | 10.20 | 12.50 | 8.98 | 8.87 | 10.60 | 13.79 | 18.96 | 19.78 | 28.65 | 20.90 | 61.16 | 0.26 | 20.88 | 47.36 | 49.92 | 22.19 |
| VitBase | | 46.86 | 47.59 | 46.88 | 42.73 | 34.21 | 50.45 | 44.7 | 56.87 | 52.63 | 56.52 | 76.06 | 31.79 | 46.72 | 65.49 | 66.03 | 51.04 |
| Tent | ✓ | 18.27 | 20.52 | 17.92 | 22.66 | 27.08 | 40.93 | 46.91 | 47.32 | 49.09 | 59.02 | 70.64 | 33.76 | 55.83 | 56.65 | 52.24 | 41.26 |
| | | 27.94 | 29.19 | 27.79 | 30.38 | 32.97 | 46.33 | 50.24 | 50.41 | 51.97 | 61.59 | 71.23 | 40.65 | 58.44 | 60.49 | 56.37 | 46.40 |
| SAR | ✓ | 13.93 | 15.71 | 13.79 | 19.66 | 24.43 | 38.68 | 45.80 | 44.28 | 47.52 | 57.74 | 70.39 | 29.37 | 54.73 | 54.48 | 49.37 | 38.66 |
| | | 20.82 | 25.80 | 22.22 | 24.78 | 29.46 | 42.47 | 49.20 | 48.91 | 49.92 | 60.12 | 70.65 | 34.74 | 57.47 | 57.93 | 53.79 | 43.22 |
| COME | ✓ | 16.47 | 18.82 | 15.87 | 21.48 | 25.68 | 39.58 | 46.21 | 45.51 | 47.75 | 57.77 | 70.32 | 31.72 | 53.90 | 54.71 | 50.19 | 39.73 |
| | | 30.89 | 32.58 | 30.80 | 33.00 | 36.21 | 48.84 | 51.55 | 53.87 | 53.09 | 62.67 | 70.56 | 44.06 | 60.00 | 61.57 | 57.27 | 48.46 |
| ETA | ✓ | 17.20 | 20.65 | 17.43 | 24.64 | 28.40 | 42.48 | 48.93 | 49.89 | 50.19 | 60.27 | 70.90 | 34.72 | 57.66 | 58.07 | 53.82 | 42.35 |
| | | 23.46 | 29.13 | 23.13 | 31.76 | 34.66 | 47.60 | 51.33 | 53.56 | 52.46 | 62.11 | 71.31 | 43.34 | 59.60 | 60.89 | 56.67 | 46.73 |
| CEMA | ✓ | 14.27 | 16.19 | 14.76 | 24.36 | 30.98 | 46.06 | 51.53 | 51.11 | 49.56 | 60.32 | 70.10 | 32.03 | 57.37 | 58.95 | 56.64 | 42.28 |
| | | 27.18 | 37.00 | 34.47 | 15.34 | 33.45 | 47.29 | 52.26 | 53.11 | 52.81 | 61.12 | 69.53 | 22.48 | 58.55 | 60.99 | 57.77 | 45.56 |

**Effect on loss functions.** We conduct a comprehensive ablation of AGR/UGR placement across the two losses (Table 16). AGR alone consistently improves accuracy; for example, enabling AGR in both losses increases performance from 38. 43% to 41. 98%. Combining AGR with UGR yields additional gains; With only one loss active, accuracy increases from 36.99% to 41.63%. The best result occurs when AGR and UGR are applied to both losses, reaching 46.02%, indicating that agreement and uncertainty are complementary and that applying them across all objectives is the most effective under TRTA.

**Effect on entropy margin.** The entropy margin m controls the confidence threshold for weighting samples based on predictive uncertainty. As shown in Figure 10, a small m imposes an overly strict threshold and downweighting informative signals from moderately confident samples, while a large margin introduces more noisy signals from unreliable predictions. Our method achieves strong performance across a broad range of values and peaks near $e$, indicating its relative insensitivity to this hyperparameter. Following the common practice in ETA (Niu et al., 2022), we set m $= 0.4 \times \log C$, which performs close to the optimum.

Table 16: Ablation on AGR and UGR. The **best** and second best results are highlighted.

| Ent | | CE | | ACC | △ |
|:---:|:---:|:---:|:---:|:---:|:---:|
| AGR | UGR | AGR | UGR | | |
| ✗ | ✗ | ✗ | ✗ | 36.99 | - |
| ✓ | ✗ | ✗ | ✗ | 38.43 | 1.44↑ |
| ✓ | ✗ | ✗ | ✓ | 42.40 | 5.41↑ |
| ✓ | ✗ | ✓ | ✗ | 41.98 | 4.99↑ |
| ✓ | ✗ | ✓ | ✓ | 45.15 | 8.16↑ |
| ✓ | ✓ | ✗ | ✗ | 41.63 | 4.64↑ |
| ✓ | ✓ | ✗ | ✓ | 45.82 | 8.83↑ |
| ✓ | ✓ | ✓ | ✗ | 42.42 | 5.43↑ |
| ✓ | ✓ | ✓ | ✓ | **46.02** | 9.03↑ |

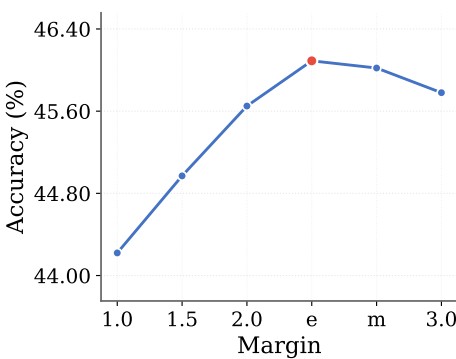

Figure 10: Ablation study on the margin.

## D  LIMITATIONS

Our proposed test-real-time adaptation (TRTA) is a new challenge in TTA, with many facets still open. While our proposed method demonstrates strong performance in the challenging TRTA setting, we acknowledge several limitations. Our reliance on an auxiliary reference model, while essential for providing a stable and high-quality adaptation process, involves considerations of computational overhead, and the base model's effectiveness is tied to the capacity of the reference model. Moreover, our work focused on image classification, extending our framework to dense prediction tasks (e.g. semantic segmentation) remains an important direction for future work.

