# OpenReview forum: "Test-Real-Time Adaptation against Sparse Knowledge Bottleneck"
_ICLR.cc/2026/Conference — ICLR 2026 Conference Withdrawn Submission_

### Official Review · Reviewer_cPF8 · 2025-10-29

**Soundness:** 2
**Presentation:** 3
**Contribution:** 2
**Rating:** 2
**Confidence:** 5

**Summary:**

This work proposes studying test-time adaptation (TTA) methods under a realistic constrained setting where the data throughput at test time matches the latency of the deployed model.
Under this setting, the stream of data reveals the next batch to the model before an adaptation step is completed leading to sparse adaptation.
Further, and to combat the performance drop of TTA methods under this sparse adaptation setting, the authors propose AUGR: Agreement and Uncertainty Guided Reweighing to complement TTA methods from the literature.
Experiments are conducted on ImageNet-C/R/K) showing performance enhancements.

**Strengths:**

The main strengths os this work are:

1) The setting of TRTA in this paper is appealing and realistic. It is important to study TTA methods under computational (and perhaps memory) constrained settings.

2) The proposed method; AUGR, is complementary and seems to be applicable to combine with existing TTA methods providing extra performance boost.

3) The paper is generally well-written and easy to understand.

**Weaknesses:**

Despite the paper's strengths, there are major weaknesses that need to be addressed before getting this paper accepted:

1) The reader gets the impression that this paper proposes the new setting "Test-Real-Time Adaptation" without citing an existing reference that concretely defined this evaluation setting and provided comprehensive evaluation under such setting: "[A]: Evaluation of Test-Time Adaptation Under Computational Time Constraints, ICML 2024". Not only the setting of TRTA is identical to the Realistic Online Evaluation Protocol in [A], even Figure 1 in the submitted paper is very similar to Figure 2 in [A]. This critical issue needs to be addressed, and significant edits in the text need to be made (e.g. removing contribution 1, adjusting the introduction, and the second paragraph of the related work).

2) While this work concerns about the computational requirements of TTA methods, it provides no discussion on the memory requirement of the proposed AUGR. In fact, AUGR requires deploying two models (instead of a single model in most TTA methods), requiring at least $2\times$ the memory consumption.

3) The definition of $k$ in line 154 is vague. Since each TTA method has a fixed $k$ (according to [A]), why would one consider evaluating each method under different $k$ (Figure 6)?

4) In the experimental setting in this work is questionable for the following reasons:

4a) Why is the reference model, in all experiments, stronger than the model that is being adapted? For example, and referencing the results in Table 1, deploying the reference model without any adaptation has a better performance than all considered TTA methods with ResNet-50. One would expect the adaptation to improve the best performing model to become better (e.g. the deployed model is the ViT, and the reference should be a small ResNet-18).

4b) The experiments reported in Table 4 are unclear. Online label imbalanced is controlled by a drichlet distribution. There is no mention on how is this distribution looking like, nor how severe is the label correlation in the stream.

4c) The experiments in Figure 6 are not realistic nor informative. Refer to weakness (3).

4d) Proper definition and mathematical formula of the predictive mutual information is missing.

4e) Experiments under continual TTA setting is missing.

4f) Ablation experiments under different batch sizes should also be included.

4f) Evaluation on ImageNet-3DCC [E]

5) Missing references:

[B] Do we really need to access the source data? source hypothesis transfer for unsuper- vised domain adaptation., ICML 2020

[C] Robust test-time adaptation in dynamic scenarios, CVPR 2023

[D] Test-Time Adaptation with Source Based Auxiliary Tasks, TMLR 2025

[E] 3d common corruptions and data augmentation. CVPR 2022

While I appreciate the orthogonality of AUGR and its consistent performance boost, the aforementioned weaknesses (especially 1 and 4) need to be addressed in order to raise my score.

**Questions:**

Please refer to the weaknesses section.

---

### Official Review · Reviewer_hfSu · 2025-10-29

**Soundness:** 1
**Presentation:** 3
**Contribution:** 2
**Rating:** 2
**Confidence:** 5

**Summary:**

Adaptation requires more computation than inference, often making it impractical to perform them simultaneously. This paper proposes a Test-Real-Time Adaptation (TRTA) setting, where inference and adaptation run in parallel. In this setting, assuming that incoming data arrive as $B_t, B_{t+1}, B_{t+2}, ... B_{t+k}, B_{2t}, B_{2t+1}, B_{2t+2}, ... B_{2t+k}, B_{3t}, ...$, the model can perform adaptation only on batches such as $B_{nt}$, while adaptation on intermediate batches $B_{nt+1}, B_{nt+2}, ... B_{nt+k}$ is not possible. The authors refer to this issue as the sparse-knowledge bottleneck. To address this, they propose AUGR, a method designed to enable effective adaptation using only $B_{nt}$. AUGR combines a multi-model ensemble strategy with uncertainty-based reweighting. The results show that AUGR can be integrated into existing TTA methods as a plug-and-play component and consistently improves their performance.

**Strengths:**

1. The paper is well-structured, making it easy to follow the proposed methodology and results.
2. The proposed TRTA setting is convincing and deserves consideration in future TTA research.
3. The visual illustrations are informative. For example, Figure 1 clearly explains the proposed setting, while Figures 2–4 effectively highlight the problems. The experimental tables and figures are also well-organized and easy to interpret.

**Weaknesses:**

1. The TRTA setting and the sparse-knowledge bottleneck are well-motivated, but the rationale behind the chosen approach is somewhat difficult to follow.

   - Their logical flow is as follows: “The model can perform adaptation only on $B_{nt}$, while adaptation on intermediate batches $B_{nt+1}, B_{nt+2}, \ldots, B_{nt+k}$ is not possible. To address this, the authors aim for effective adaptation using only $B_{nt}$.”

   - However, the following directions seem more reasonable: (1) “To address this, propose a more efficient method that can perform faster adaptation,” or (2) “To address this, propose a method that prevents knowledge loss from $B_{nt+1}, B_{nt+2}, \ldots, B_{nt+k}$.”

   - The direction selected in the paper is not fully convincing. Specifically, even if the proposed method achieves effective adaptation using only $B_{nt}$ (as described in Line 86: learning primary signal from $B_{nt}$), it does not mean that knowledge from $B_{nt+1}, B_{nt+2}, \ldots, B_{nt+k}$ can be captured. Therefore, it is questionable whether the sparse-knowledge bottleneck is truly resolved.

   - In other words, the connection between their proposed method and the emphasized problem needs to be made stronger.

2. It would be valuable to emphasize more clearly what is novel about their proposed methodology.

   1. Agreement-guided reweighting leverages the consensus between multiple models (inference and reference). Although there may be subtle differences, such ensemble-based strategies are widely used. For example, previous TTA methods have employed consensus of teacher(ie, reference model)–student or multiple teacher's decision [1~4, 6].
   2. Uncertainty-guided reweighting adjusts the strength of adaptation based on entropy. This idea has been explored in prior work. For instance, EATA [5] states that “Our weighting function excludes high-entropy samples from adaptation and assigns higher weights to test samples with lower prediction uncertainties, allowing them to contribute more to model updates.” It is difficult to identify what new insight the proposed method provides.

3. Additional evidence is needed to show that the improvement does not come from leveraging the strong and robust performance of the reference model.
   1. In Table 1, ViTBase shows robust performance, while ResNet-50 performs poorly. Existing methods such as TENT and SAR are built on ResNet-50, which naturally leads to lower results, whereas the proposed method uses ViTBase, making the performance gain rather expected.
   2. It is unclear whether the proposed method would remain effective if the reference model had performance comparable to or even lower than ResNet-50 (e.g., WideResNet).
   3. Moreover, the proposed framework requires storing three models, the inference model, the adapting model, and the reference model, which is memory-inefficient for real-world applications.
4. Some ambiguous expressions in the paper need to be clarified. The reviewer has the following questions:

   1. Line 79: Could the authors clarify what is meant by ‘often reduce compatibility’? Techniques such as quantization or parameter update control [7] can also make existing methods faster for adaptation.
   2. Lines 85 and 164: Could the authors clarify what is meant by ‘make adaptation sensitive to noise’? A clearer explanation might be: “The delay prevents immediate updates, hinders rapid adaptation to the target domain, and makes the model vulnerable to sudden environmental changes.”
   3. Several other vague expressions should be revised.

[1] Continual Test-Time Domain Adaptation CVPR 2022

[2] A Probabilistic Framework for Lifelong Test-Time Adaptation CVPR 2023

[3] Persistent Test-time Adaptation in Recurring Testing Scenarios NeurIPS 2024

[4] Improved Self-Training for Test-Time Adaptation CVPR 2024

[5] Efficient Test-Time Model Adaptation without Forgetting ICML 2022

[6] MM-TTA: Multi-Modal Test-Time Adaptation for 3D Semantic Segmentation CVPR 2022

[7] EcoTTA: Memory-Efficient Continual Test-time Adaptation via Self-distilled Regularization CVPR 2023

**Questions:**

Please refer to the weaknesses above.

---

### Official Review · Reviewer_hVc2 · 2025-10-31

**Soundness:** 2
**Presentation:** 3
**Contribution:** 2
**Rating:** 4
**Confidence:** 3

**Summary:**

This paper introduces **Test-Real-Time Adaptation (TRTA)**, a new paradigm for test-time adaptation (TTA) that enables simultaneous prediction and model adaptation without synchronization pauses. The core challenge addressed is the **sparse-knowledge bottleneck**: in real-time scenarios, adaptation runs concurrently with inference but completes more slowly, resulting in far fewer model updates than predictions. This scarcity of updates prevents effective error correction and reliable knowledge accumulation.

To address this bottleneck, the authors propose **Agreement- and Uncertainty-Guided Reweighting (AUGR)**, which combines two complementary signals: (i) **inter-model agreement** between a base model and a reference model (measured via Top-$K$ rank concordance and Jaccard overlap), which identifies common knowledge robust to noise; and (ii) **inner-model uncertainty** (entropy-based weighting), which down-weights low-confidence predictions. AUGR is designed as a plug-and-play module applicable to existing TTA methods. Experiments on ImageNet-C/R/K demonstrate improvements in terms of average accuracy with consistent gains across various baselines.

**Strengths:**

### Originality
- **Novel problem formulation**: The paper identifies and formalizes the TRTA paradigm, which is practically motivated and distinct from standard TTA. The distinction between predict-then-adapt (TTA) and predict-and-adapt (TRTA) is clear and relevant to real-world applications like autonomous driving and video analytics.
- **Sparse-knowledge bottleneck**: The characterization of the sparse-knowledge bottleneck via Predictive Mutual Information (PMI) is insightful and provides a principled way to understand why conventional TTA methods fail under real-time constraints.
- **Complementary design**: The combination of agreement-guided reweighting (AGR) and uncertainty-guided reweighting (UGR) is well-motivated. The authors provide empirical evidence (Figure 3, Figure 4) showing why both signals are necessary and how they complement each other.

### Quality
- **Comprehensive experiments**: The experimental evaluation covers multiple benchmarks (ImageNet-C/R/K), different corruption types, severity levels, mixed distribution shifts, and imbalanced label distributions. The ablation studies are thorough, examining key components, hyperparameters, adaptation delays, and model pairs.
- **Consistent improvements**: AUGR shows consistent gains across five baseline methods (Tent, SAR, COME, ETA, CEMA), demonstrating its general applicability.
- **Practical relevance**: The paper addresses a real deployment constraint (real-time inference) that is often overlooked in the TTA literature.

### Clarity
- The paper is generally well-written and well-structured. The problem is clearly motivated, the method is explained step-by-step, and the experimental section is comprehensive.
- Figures 1, 2, and 3 effectively illustrate the core concepts (TRTA paradigm, PMI trends, complementarity of AGR and UGR).
- Algorithms 1 and 2 provide pseudocode that clarifies the implementation.

### Significance
- The TRTA paradigm opens a new research direction in test-time adaptation, bringing the field closer to real-world deployment requirements.
- AUGR's plug-and-play nature makes it easy to integrate with existing methods, which enhances its practical value.

**Weaknesses:**

### 1. **Limited Novelty in Method Design**
While the problem formulation (TRTA) is novel, the proposed solution (AUGR) is **relatively incremental**. The core idea of using inter-model agreement and entropy-based uncertainty is well-established in the literature:
- **Agreement-based learning** has been extensively studied in co-training, noisy label learning, and domain adaptation (e.g., Li & Hoiem 2017, Zhang et al. 2018, Han et al. 2018).
- **Entropy minimization and uncertainty weighting** are standard in TTA (e.g., ETA, SAR, COME).

The main contribution is the **combination** of these ideas in the TRTA setting, but the method itself lacks significant technical novelty. The authors do not propose fundamentally new mechanisms; rather, they adapt existing techniques to a new setting.

**Specific concerns**:
- The rank-alignment score (Eq. 5) and Jaccard index (Eq. 6) are simple heuristics. Why is this particular combination optimal? Have the authors considered other agreement metrics (e.g., KL divergence, Wasserstein distance, or cosine similarity on logits)?
- The entropy-based weighting (Eq. 8) is standard. The only difference from prior work is the use of **soft weighting** instead of **hard filtering**, but this is a minor modification.

### 2. **Computational Overhead and Practicality**
The paper claims TRTA is suitable for real-time deployment, but the method **requires a reference model** (e.g., VitBase), which incurs additional computational cost:
- **Inference overhead**: The reference model requires an extra forward pass for every batch, which increases latency and memory usage.
- **Storage overhead**: Maintaining two models (base and reference) increases memory requirements, which may be prohibitive on edge devices.

The authors mention that "a historical version of the base model" can serve as the reference (Table 7), but this ablation shows **lower performance** compared to using a frozen foundation model (Table 1). This suggests that the quality of the reference model is critical, but the paper does not provide clear guidance on how to select or maintain a good reference model in practice.

**Missing analysis**:
- The paper does not report **wall-clock time** or **memory usage** for AUGR in Tables 1-4. While Table 9 compares training-free methods, it does not include AUGR's overhead relative to standard TTA.
- For edge-cloud scenarios, the paper assumes the reference model is hosted in the cloud, but this introduces **communication latency**, which is not addressed.

### 3. **Weak Theoretical Justification**
The paper lacks theoretical analysis of why AUGR mitigates the sparse-knowledge bottleneck. The method is motivated by intuitive arguments and empirical observations (Figure 3, Figure 4), but there is no formal analysis of:
- **Convergence guarantees**: Does AUGR ensure that the model converges to a good solution under sparse updates?
- **Sample complexity**: How many updates are required for AUGR to achieve a certain level of performance?
- **Robustness**: Under what conditions does AUGR fail? When does agreement mislead the model (e.g., when both models are wrong)?

The lack of theory makes it difficult to predict when AUGR will work well and when it might struggle.

### 4. **Incomplete Experimental Analysis**

#### (a) **Limited Baseline Comparisons**
The paper compares AUGR against five TTA baselines (Tent, SAR, COME, ETA, CEMA) and three training-free methods (LAME, T3A, FOA). However, several important recent TTA methods are missing:
- **EATA** (Niu et al., 2022): A more recent sample-selection-based method.
- **NOTE** (Gong et al., 2022): A normalization-based TTA method.
- **RoTTA** (Yuan et al., 2023): Explicitly designed for robust TTA in dynamic scenarios.
- **AdaContrast** (Chen et al., 2022): A contrastive TTA method.

Without comparisons to these methods, it is unclear whether AUGR's improvements are state-of-the-art or merely competitive with older baselines.

#### (b) **Insufficient Analysis of Agreement Failure Cases**
The paper assumes that **agreement implies correctness**, but this is not always true. Figure 3 shows that Area 2 (high agreement, high certainty) has 84.6% accuracy, which means **15.4% of high-agreement samples are still incorrect**. The paper does not analyze:
- What causes the reference and base models to agree on wrong predictions?
- How does AUGR perform when both models are biased in the same direction (e.g., both predict the same wrong class with high confidence)?

#### (c) **Limited Analysis of Adaptation Delay $k$**
Figure 6 and Tables 9-12 show that performance degrades as $k$ increases, but the analysis is shallow:
- What is the **critical value** of $k$ beyond which AUGR becomes ineffective?
- How does $k$ relate to the **distribution shift severity**? Intuitively, stronger shifts should require more frequent updates, but the paper does not explore this.

#### (d) **Missing Comparisons with Asynchronous Training Literature**
The paper claims TRTA is related to asynchronous inference and training (Section 2), but it does not compare AUGR with methods from this literature (e.g., edge-cloud collaborative learning, online continual learning). This is a missed opportunity to position TRTA within a broader context.

### 5. **Hyperparameter Sensitivity**
While Figure 8 shows ablations on $\alpha$, $\lambda_a$, $\lambda_u$, and Top-$K$, the paper does not provide clear guidance on how to set these hyperparameters in practice:
- The optimal values appear to be **dataset-dependent** (e.g., $m = 0.4 \times \log C$ is borrowed from ETA, but is this optimal for TRTA?).
- The paper does not discuss how to **tune these hyperparameters** when labels are unavailable at test time (which is the whole point of TTA).

### 6. **Limited Discussion of Limitations**
The paper acknowledges that AUGR is limited to image classification and that extending to dense prediction tasks (e.g., semantic segmentation) is future work. However, it does not discuss:
- **Failure modes**: Under what conditions does AUGR fail? When does agreement mislead the model?
- **Scalability**: How does AUGR scale to larger models (e.g., CLIP, DINO, or LLaMA-style vision-language models)?
- **Non-IID data streams**: The experiments assume IID or temporally correlated data, but real-world streams may exhibit sudden distribution shifts. How does AUGR handle such cases?

### 7. **Writing and Presentation Issues**
Despite being generally well-written, the paper has some issues:
- **Verbose explanations**: Some sections (e.g., Section 3.2.1) are overly detailed, while others (e.g., the relationship between PMI and knowledge accumulation) are underdeveloped.
- **Missing details**: The paper does not specify how the reference model is updated (if at all). Is it frozen, or does it adapt alongside the base model?

**Questions:**

1. **Agreement metrics**: Why do you use Top-$K$ rank alignment and Jaccard index? Have you tried other agreement metrics (e.g., KL divergence, Bhattacharyya distance, or cosine similarity on logits)? How sensitive is AUGR to the choice of agreement metric?

2. **Reference model selection**: How should practitioners choose the reference model in real-world scenarios? What are the trade-offs between using a frozen foundation model, a historical checkpoint, or an ensemble?

3. **Failure cases**: Can you provide examples where AUGR fails? For instance, when both models agree on the wrong prediction, does AUGR amplify the error? How can this be mitigated?

5. **Computational cost**: Can you report the **wall-clock time and memory usage** of AUGR compared to baseline methods? How much overhead does the reference model introduce?

6. **Adaptation delay $k$**: What is the **critical value** of $k$ beyond which AUGR becomes ineffective? How does this depend on the severity of the distribution shift?

7. **Hyperparameter tuning**: How do you recommend setting $\alpha, \lambda_a, \lambda_u$, and Top-$K$ in practice when labels are unavailable? Can these be adapted online?

8. **Comparison with asynchronous training**: How does AUGR compare with methods from the edge-cloud collaborative learning and online continual learning literature? Can AUGR be combined with these methods?

9. **Extension to dense prediction**: Can AUGR be extended to semantic segmentation or object detection? What modifications would be required?

10. **Non-IID data streams**: How does AUGR handle sudden distribution shifts (e.g., from sunny to rainy weather in autonomous driving)? Does the agreement signal degrade rapidly in such cases?

11. **Comparison in other realistic setting:** Moreover, the robustness of the proposed approach can be demonstrated if it performs well in the continual TTA setting as in **CoTTA** (Wang et al., 2022) and **PETAL** (Brahma et al., 2023).

---

### Official Review · Reviewer_Ag7J · 2025-11-01

**Soundness:** 2
**Presentation:** 2
**Contribution:** 2
**Rating:** 4
**Confidence:** 3

**Summary:**

This paper introduces Test-Real-Time Adaptation (TRTA), a novel paradigm for test-time adaptation where inference and adaptation proceed in parallel without synchronization waits, unlike traditional TTA, which pauses inference for adaptation. The key innovation is identifying and addressing the "sparse-knowledge bottleneck" - when adaptation opportunities are limited due to parallel processing, knowledge accumulation stalls. The authors propose Agreement- and Uncertainty-Guided Reweighting (AUGR), which combines inter-model agreement (measuring concordance of predicted class rankings between base and reference models) with inner-model uncertainty (evaluating reliability by down-weighting low-confidence predictions). Experiments on ImageNet-C/R/K demonstrate improvements of 9.03%, 3.88%, and 5.50% respectively, and AUGR functions as a plug-and-play module compatible with existing TTA methods (Tent, SAR, COME, ETA, CEMA).

**Strengths:**

**Novel Problem Formulation**: TRTA addresses real-world latency constraints ignored in prior TTA work. The predict-and-adapt paradigm is directly applicable to autonomous driving, video analytics, and edge computing, where inference pauses are unacceptable.

**Clear Problem Analysis**: PMI analysis on ImageNet-C quantitatively shows knowledge accumulation stalls under TRTA (PMI curve flattens vs. steady growth in TTA), particularly as adaptation delay k increases from 3 to 50 batches.

**Comprehensive Experiments**: Evaluation spans multiple benchmarks (ImageNet-C/R/K), architectures (ResNet-18/50-GN, ViT-Base, EfficientViT-B1), and five baselines. Notably recovers CEMA's failed cases (e.g., "Elastic": 0.36%→46.22%).

**Weaknesses:**

**Computational Overhead Unquantified**: Requires auxiliary reference model but lacks: (a) memory footprint analysis for edge deployment, (b) wall-clock time vs. traditional TTA, (c) energy consumption measurements, (d) actual FPS on standard hardware. Figure 9 shows that some training-free methods are slow but do not quantify AUGR's own overhead. Claims of "realistic for real-time" are unsubstantiated without these metrics.

**Reference Model Dependency**: Authors acknowledge effectiveness "tied to reference model capacity" but don't address: (a) behavior when reference model degrades under distribution shift, (b) systematic selection guidelines for practitioners, (c) performance patterns with large capacity gaps (Tables 13-15 show results without analysis), (d) whether domain-specific pre-training is needed. This dependency complicates deployment with unclear model selection criteria.

**Questions:**

**Computational Cost**: Provide wall-clock time (ms/batch) and memory footprint on edge hardware comparing direct inference, traditional TTA, and TRTA+AUGR?

**Reference Model Selection**: What systematic guidelines should practitioners follow to select reference models for new domains?

---

### Official Review · Reviewer_6x4y · 2025-11-05

**Soundness:** 2
**Presentation:** 3
**Contribution:** 1
**Rating:** 2
**Confidence:** 5

**Summary:**

This paper introduces the Test-Real-Time Adaptation (TRTA) paradigm as an alternative to Test-Time Adaptation (TTA), and proposes the Agreement- and Uncertainty-Guided Reweighting (AUGR) method for TRTA. As opposed to TTA, which poses "adaptation delays due to self-training", TRTA creates a reference copy of the base model in order to perform simultaneous inference (forward pass) and adaptation (backward pass). However, to constrain the additional cost incurred by backward passes, the authors use sparse backprop updates, which creates a sparse knowledge bottleneck problem, since training signals are rare (hence error correction and knowledge accumulation suffers for TRTA relative to TTA). To mitigate this, the authors introduce AUGR, which measures the inter-model agreement (between base and reference models) on top-K class rankings, and the inner-model uncertainty (promoting reliable signals and penalising low-confidence cases). This is implemented via a weighted sum of two terms: a Jaccard index score (ranked class alignment) and entropy-based margin score. Experimentally, the authors compare their proposed AUGR with TTA baseline methods such as Tent, ETA and others. Tests are run on the ImageNet-C/R/K datasets with ResNet-50 and ViTBase models. The authors report that AUGR most consistently resolves the sparse knowledge bottleneck problem on a variety of TRTA tasks (mixed distribution shifts and online imbalance label distribution shifts).

**Strengths:**

**S1 [Writing]** – This paper is very coherently written and easy to follow. Exposition is clear, claims are built up logically and well-organised into highly comprehensible sections.

**S2 [Presentation]** – The presentation is noteworthy and claims (on AUGR's competitiveness in TRTA) are substantiated with numerical evaluations, qualitative results, diagram-form/algorithmic explanations.

**S3 [Reproducibility]** – The authors are very clear on the experimental setup, hyperparameter selection and ablation studies performed. They also state that "code will be released", which inspires confidence that this submission contains reproducible results.

**Weaknesses:**

**W1 [Significance]** – The reviewer is not adequately convinced about the proposed TRTA paradigm. It appears to rely on cloning out an additional reference model (serious memory and communication overhead); it appears to perform sparse updates (hence inference and adaptation cannot be truly "simultaneous" w.r.t. the entire dataset); it appears to pose additional and complex problems (the "sparse knowledge bottleneck" problem) which existing paradigms do not suffer from.

**W2 [Originality]** – The idea of simultaneously training on inference-time data is well-researched in meta learning, continual learning, active learning and self-knowledge distillation literature. This manuscript has not engaged with notable prior ideas, neither conceptually or experimentally.

**W3 [Quality]** – In the age of large vision-language models, the reviewer finds it insufficient to evaluate only ResNet-50 and ViT-Base model architectures, since they are relatively small, less in use; results on ResNet-50 are not informative about performance on frontier models, especially given the memory/communication overhead of the TRTA paradigm.

**Questions:**

**Q1** – Could the authors justify further the TRTA setting? The reviewer acknowledges that TTA is not simultaneous (in inference and adaptation) and dense in its updates, but TRTA employs sparse updates, hence cannot truly realise full simultaneous adaptation and inference for all data points in an inference set either.

**Q2** – Have the authors compared AUGR against common baselines such as LoRA adaptation, projected gradient descent or linear probing?

**Q3** – Have the authors considered the additional space overhead incurred by copying out (and storing) a reference model? This would be a severe limiting factor for frontier models, where memory footprint is a primary concern.

---

### Note · Authors · 2025-11-12

I have read and agree with the venue's withdrawal policy on behalf of myself and my co-authors.